# Fast imaging of millimeter-scale areas with beam deflection transmission electron microscopy

Zhihao Zheng [1], Christopher S. Own[2], Adrian A. Wanner[1,3], Randal A. Koene [2], Eric W. Hammerschmith[1], William M. Silversmith[1], Nico Kemnitz [1], Ran Lu[1], David W. Tank [1] & H. Sebastian Seung [1] ✉

Serial section transmission electron microscopy (TEM) has proven to be one of the leading methods for millimeter-scale 3D imaging of brain tissues at nanoscale resolution. It is important to further improve imaging efficiency to acquire larger and more brain volumes. We report here a threefold increase in the speed of TEM by using a beam deflecting mechanism to enable highly efficient acquisition of multiple image tiles (nine) for each motion of the mechanical stage. For millimeter-scale areas, the duty cycle of imaging doubles to more than 30%, yielding a net average imaging rate of 0.3 gigapixels per second. If fully utilized, an array of four beam deflection TEMs should be capable of imaging a dataset of cubic millimeter scale in five weeks.

Volume electron microscopy (vEM) is so far the only approach that has ever been used to reconstruct a connectome—a complete map of neural connectivity at synaptic resolution for an entire brain or nervous system[1–4]. In general, vEM can be categorized into scanning electron microscopy (SEM) or transmission electron microscopy (TEM) techniques[5,6], depending on whether the electron beam is raster-scanned over a sample surface or passed through a thin section of the sample. Block face imaging using single-beam SEMs[7,8] has been widely adopted and in the case of focused ion beam SEM (FIB-SEM)[9–11], offers superior axial resolution. However, single-beam imaging, which sequentially scans one pixel at a time, is inefficient and limits the size of the imaged volume. Recent developments of multiple parallel electron beams have dramatically increased throughput, achieving a burst imaging rate of up to 1.8 gigapixel per second (50 ns dwell time) on brain tissues for a 91-beam SEM[12–16]. However, in both SEM and TEM approaches significant overhead is incurred by non-imaging tasks, such as stage motion and stabilization, focus adjustments, and system maintenance. For example, recently the effective imaging rate of acquiring a large dataset with a 61-beam SEM was reported to be 0.19 gigapixel per second (200 ns dwell time)[17].

In principle, TEM has superior imaging efficiency as all electrons are captured simultaneously. However, transmission imaging requires the tissue block to be sliced into consecutive ultra-thin sections, a technique known as serial section TEM (ssTEM). Classically, ssTEM is used for 3D reconstruction of neural tissues at small scale[18–20]. At large scale, cutting and collecting tens of thousands of ultra-thin sections consistently is challenging. Large-scale imaging with TEM is therefore either laborious or necessitates custom engineering for automation. In the 1980s, ssTEM was used for mapping the connectome of *C. elegans*[1]. Later on, ssTEM was used to acquire 3D EM datasets of multiple *C. elegans*[21,22], an entire fly larval brain[23], normal and pathological retinas[24,25], and mouse cortex[26,27]. Recently, automation and parallelization of TEM[26,28,29] have allowed imaging of ever larger volumes, such as a complete adult fly brain[2] and a cubic millimeter volume of mammalian cortex[30].

The previous state of the art in high throughput TEM was a system at the Allen Institute for Brain Science[28]. This achieves high imaging speed through several innovations. A reel-to-reel tape translation system allows automated delivery of sections based on GridTape technology[29]. A CMOS camera with a 30-megapixel effective frame size and a low distortion lens achieves a burst imaging rate of 0.5 gigapixels per sec (GPix/s). A scalable software infrastructure allows closed-loop workflow management based on real-time image processing[31]. However, after factoring in both imaging and non-imaging overheads like stage movement and microscope downtime, it still required six months to image a cubic millimeter volume with six TEMs[28]. Furthermore, there is a growing need to image much larger datasets[3,32].

[1]Princeton Neuroscience Institute, Princeton University, Princeton, NJ, USA. [2]Voxa, Seattle, WA, USA. [3]Present address: Paul Scherrer Institute, Villigen, Switzerland. ✉e-mail: sseung@princeton.edu

Constrained by microscope optics, TEM has a relatively small field of view. To image a square millimeter area, it requires thousands of x-y stage translations to acquire image tiles that are later stitched together to form a multi-tile image. Because the time to image a tile has become so fast the stage translations require more time than the image capture itself[28]. A massive number of positional translations—typically ~10,000 per mm$^2$—also rapidly consumes piezo stage life of ~10 billion cycles of the crystal per year when operated 24/7. Therefore, to advance the state of the art, it is essential to reduce the overhead due to stage translations.

Deflection of electron beams is widely used in lithography and microscopy to avoid unintended exposure of samples[33–35]. The control and behavior of beam deflection have been carefully examined down to picosecond time scale[36–38]. In comparison, stage movement is often followed by a residual drift and requires a settling time of tens of milliseconds[2,39]. Beam deflection is therefore commonly used to minimize overhead and side effects of stage motion. In structural biology, to capture images of the same structure at many different orientations, beam deflection is used to take a large number of particle images, each at a different, previously unexposed location, all without moving the stage[40–42]. However, to image a large area at millimeter scale and nanoscale resolution, there's a limitation of how far beam deflection can extend. While existing software can combine beam deflection and stage movement to image a large area[43,44], the performance is not sufficient for high-throughput imaging.

## Results

A beam deflection mechanism (Cricket™) for high-throughput TEM imaging has been proposed briefly[28]. Here we report the development of a TEM imaging pipeline that integrates beam deflection with state-of-the-art stage, section feed, and camera technologies (Supplementary Fig. 1a), including an automated reel-to-reel tape translation system[28,29] and a CMOS camera with a 36-megapixel frame size[28]. We present quantifications of several key aspects of the system, including imaging speed, quality, capacity, and its application across resolutions. The bdTEM achieves the fastest TEM imaging rate to date, a threefold increase over the previous TEM with the same camera[28].

### Beam deflection (Cricket)

We repurposed electromagnetic lens deflectors in the TEMs (Fig. 1a, b). The deflectors above the object plane shift the electron beam over the specimen and simultaneously the deflectors below the object plane precisely de-scans the image back onto the TEM camera. As a result, multiple image tiles (a supertile) can be acquired without stage movement (Fig. 1c), a significant improvement over regular TEMs limited to 1 image tile per stage position. While an arbitrary beam deflection sequence can be used, we opted for a spiral arrangement beginning from the center non-deflected tile because the first tile has the best image quality. When a matrix of 3 × 3 tiles were used for large-scale imaging, overhead for stage motion is significantly reduced, accounting for only 4% of section imaging time with an additional 6% of Cricket settling time (Fig. 1e, Quantification of imaging rates in Supplementary Methods). As a result, the proportion of time spent on image acquisition out of total time per section, referred to as the imaging duty cycle, is greatly increased. The imaging duty cycle is 32.9% for 3 nm per pixel, and 31% for 4 nm per pixel (Methods), a twofold increase compared to a previous peak performance of 15%[28].

### Capacity and image quality of beam deflection

We explore the limit of beam deflection by attempting to acquire a maximal number of subtiles at a magnification of 4 nm per pixel, ensuring that each subtile has sufficient overlap with its four neighboring subtiles for seamless montaging. Notably, a supertile maximally consists of 30 full-size subtiles in a rectangle grid of 6 by 5, or 44

subtiles in a cruciform shape (Fig. 2a), with a size of 36 megapixels per subtile (6000 × 6000). At a pixel size of 4 nm, the stitched supertile covers an area exceeding 100 μm × 100 μm after subtile overlaps are taken into account (Fig. 2b). The image tiles at the outermost edge are not taken into account because they either are significantly obstructed by a circular boundary within the microscope, or have poor quality due to low beam intensity. Mechanical adjustment of objective aperture didn't affect the extent of the boundary, indicating that the compromised beam intensity is likely to be caused by other microscope components—for example, the objective-lens pole piece. Beam deflection can additionally be used across a range of magnifications, up to 100,000 times with a pixel size of 0.09 nm (Supplementary Fig. 2).

Zoom-in images show that beam deflection causes a reduction in image quality (Fig. 2c). We therefore used signal-to-noise ratio (SNR) as a function of spatial frequency[45] to assess image quality across subtiles quantitatively (Methods). Beam deflection causes a slight decrease of SNRs for tiles located farther away from the center (Fig. 2d, Supplementary Fig. 1b). Evaluation of image deformation after stitching reveals distinct deformation fields for each subtiles, with a maximum distortion of approximately 20 pixels within a 6000 × 6000 pixel tile (Fig. 2e). The beam deflected subtiles have slightly larger distortion than the center tile. Here factors that affect image quality may include defocus[36], increasing aberration[41], or reduced electron dose. Importantly, the minor reduction of SNRs for 8 deflected tiles in the 3 × 3 configuration have no effect on image stitching (Fig. 3a–f), alignment, and segmentation (Fig. 4, Supplementary movie 1-2).

### Stage and section transition

The TEMs are integrated with a two-axis piezo-driven stage and a previously reported reel-to-reel tape translation system[28,29]. In previous volume TEM techniques, stage step-and-settle time ranged from under 50 ms to 100 ms for one axis[2,28]. The stage in our TEMs achieves an average motion time of 34 ms and 40 ms respectively on x and y axes (Supplementary Fig. 1c, d), increasing efficiency on the remaining stage motions.

The reel-to-reel tape translation system allows automated section exchange and barcode-based slot identification using GridTape technology[28,29]. GridTape is a tape substrate that has regularly spaced apertures (i.e. slots), each milled with an unique barcode and covered with an electron-translucent film to provide support for a section. The transition to image the next section includes: (1) advance the tape to the next slot based on the barcode; (2) capture low-magnification overview image, which is used to define region-of-interest for subsequent high-resolution imaging; (3) center the electron beam, which could be off center due to lens hysteresis; (4) find the best focus for the current section. Basic software has been implemented to run these steps automatically (Supplementary Figs. 5, 6; Supplementary movie 3). With improved imaging efficiency and significantly reduced stage motions, the overhead associated with these section transition steps now constitutes a substantial contribution of the total time, at ~35% (Fig. 1d, Table 1).

### Optics

The detector optics of the TEM is upgraded with a compact light-optical lens assembly and a high speed CMOS camera that yields a frame size of 6000 × 6000 pixels (or 36 MPix). The sensor sits near the film plane of a conventional TEM (Supplementary Fig. 1a), eliminating the impractical need of an extended vacuum column required by the past generations of large pixel-count detectors[26,28]. The initial dataset was acquired at an exposure time of 120 ms (Fig. 3a–e, Supplementary movie 1). Images acquired at an exposure time of 40 ms have lower SNR but sufficient resolution for synapse identification and reconstruction of neuronal morphology (Fig. 3f, g, Supplementary movie 2). As a result, the burst imaging rate (0.9

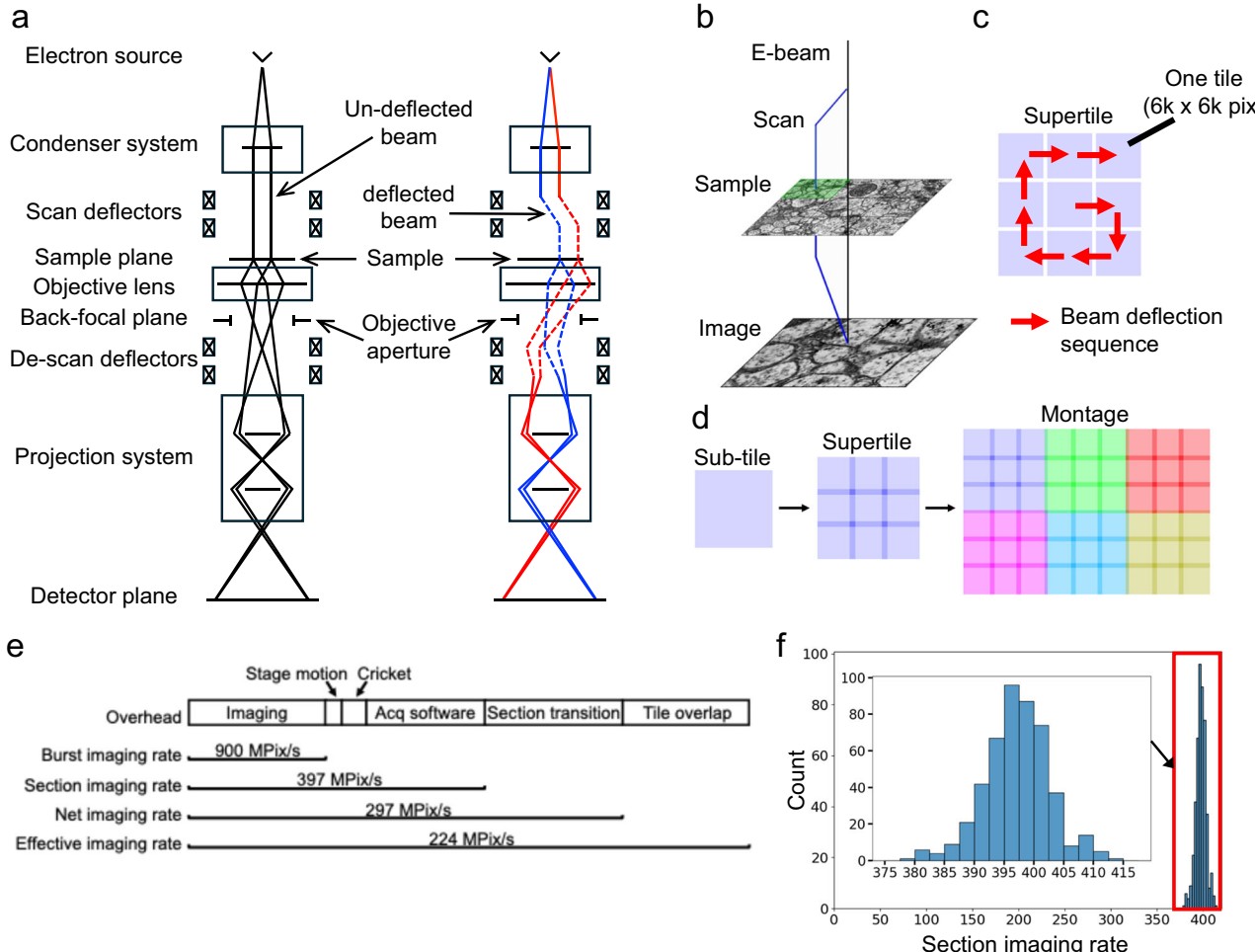

**Fig. 1 | Beam deflection for TEM imaging. a** Ray diagram describing the optical paths for a central (left) and a singly deflected (right) subtile position in bd-TEM. A pair of rays propagating from different points of a finite electron source are formed into a parallel beam on the sample, and are forward propagated from the sample and formed into a projected and magnified image on the detector (camera) plane. The beam is deflected above the sample to illuminate an adjacent location on the sample, and the subsequent image is returned to the optic axis to be acquired by the camera. **b–d** Cricket sequentially scans each tile of a 3 × 3 matrix pattern on the sample, producing a "supertile" at a stage position with overlap between tiles. Neighboring supertiles (different colors) can be stitched together to form a mon-tage image. **e** Imaging speed based on acquisitions at 3 nm/pix (Supplementary Methods). The bar lengths are proportional to the contribution of overheads to imaging speed. Computational overhead of acquisition ("acq software") includes communications between different components, software time jitters, computing image statistics, and writing image files. Effective imaging rates are calculated from the total number of pixels of a montage acquisition with overlap pixels only counted once ("tile overlap"), over the same overhead as in the net imaging rate. Stage movement and Cricket settling times are determined by the number of tiles and generally remain constant for a specific section (Supplementary Methods). Therefore, the variability in their proportions is determined by the variability in section imaging rate (-1.4% **f**). **f** The distribution of section imaging rates for 472 sections. The size of each section is 1.9 × 1.2 mm². The average rate is 397.1 MPix/s (s.d. 5.5).

GPix/s, imaging only) of bdTEMs is 1.8 times that of the previous fastest imaging rate[28] (0.5 GPix/s).

## Quantification of imaging speed

Overall, these modifications dramatically increase imaging through-put. Acquisition of a 1 mm² section takes 8.8 minutes including ima-ging, stage motion, and section transition (net average imaging rate of 0.3 GPix/s at 3 nm/pix, Table 1, and Supplementary movie 3). The net imaging rate is 3 times the peak imaging rate of the previous fastest TEM at 0.1 GPix/s with the same camera[28]. The previous imaging rate[28] includes overhead for image quality control and post-processing. These functionalities have not yet been integrated into our acquisition software but in principle could be performed during the overhead for acquisition software and section transition. To understand the varia-tion of imaging speeds over continuous imaging of a large number of sections, we measured the section imaging rates across 472 sections in a series (Fig. 1f). The standard deviation of imaging speed is minimal, at

-1.4% of per-section speed, demonstrating the consistency of the TEM system.

## Automated serial sectioning

The bdTEM imaging system is equipped with reel-to-reel section feed mechanism, which requires serial ultrathin sections to be cut and collected on GridTape. Previous efforts have modified an Automated Tape-Collecting Ultramicrotome (ATUM)[46] to accommodate the reg-ularly spaced apertures and delicate films in GridTape[28,29]. We have similarly modified an ATUMtome (RMC/Boeckeler) to be compatible with an ultramicrotome (Leica UC7) and GridTape (Supplementary Fig. 3). The hardware and software modifications include: (1) reposi-tioning feed and take-up motors to accommodate the large-size GridTape reel, (2) adding a high-precision, three-axis motorized stage, (3) adding a before-sectioning and an after-sectioning cameras to monitor tape status, in addition to the UC7 camera module, (4) implementing closed-loop control software to synchronize tape

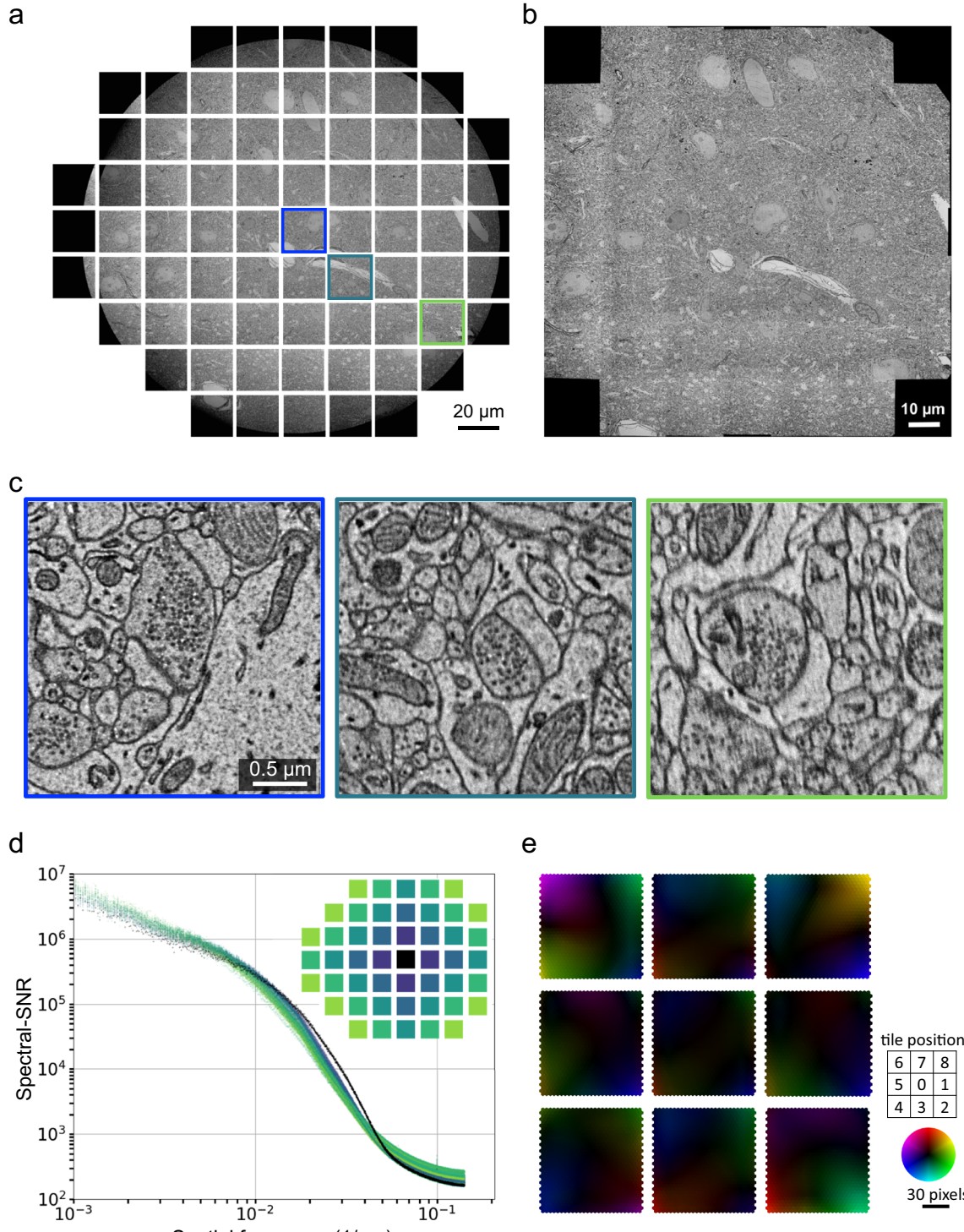

**Fig. 2 | Limit and image quality of beam deflection. a** Maximal number of subtiles imaged with beam deflection in the JEOL 1200EX-II TEM. Each tile has 6000 × 6000 pixels with a pixel size of 3.6 nm. Representative images of 3 independent experiments. **b** A fully stitched montage of the full subtiles in **a**. Each tile and its neighboring tiles have overlap, which is used for the stitching. **c** Zoom-in images of three highlighted tiles from **a**. **d** The signal-to-noise ratio (SNR) as a function of spatial frequency (Methods) for each image was computed and shown with colors of the tiles indicated in the legend. Spectral-SNR is the integral over the full spectral width. **e** Distortion of the 9-tile configuration. The non-rigid displacements (Methods) of evenly spaced image features before and after stitching are visualized as color-intensity-encoded vectors. Color hue indicates the direction and intensity indicates the magnitude of displacement at a scale of 30 pixels (radius of color wheel).

movement with cutting, (5) implementing feedback control of water level based on a proportional-integral-derivative principle. After these modifications, we have completed two successful serial sectioning and collection of over 3000 and 9000 sections, respectively.

## Montaging and reconstruction

We set out to test whether images acquired by the bdTEMs can be montaged into millimeter-size sections and subsequently reconstructed into a 3D EM volume. A low magnification (50×) overview image of the section was taken to extract ROI for high-resolution

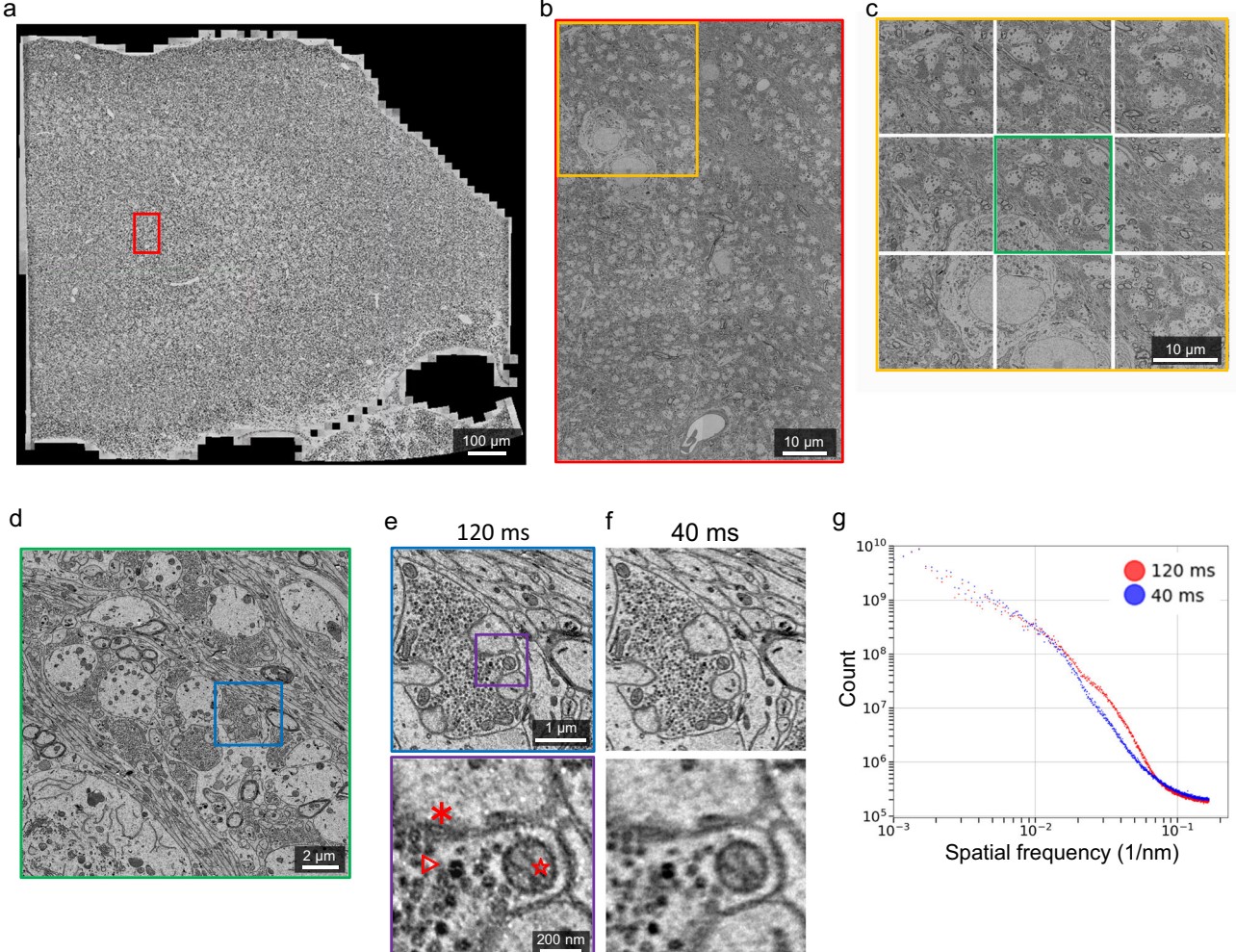

**Fig. 3 | High-throughput millimeter-area imaging with beam deflection. a** A fully stitched montage of a hippocampus section (1 mm²) imaged with a Cricket-equipped TEM, at a pixel size of 3 nm. The montage includes 4320 tiles (or 480 supertiles). Representative data of over 3000 sections. **b** A montage of 6 stitched supertiles (from red outline in **a**). **c** A Cricket supertile consisting of 9 tiles (from orange outline in **b**). Each tile has 6000 × 6000 pixels and the overlap between neighboring subtiles of a supertile is 15% (~900 pixels). **d** A single image tile (from green outline in **c**). **e** Further zoom-in shows postsynaptic density (\*), synaptic vesicles (△) and a cross-section plane of a mitochondria (☆). **f** Same area and zoom-in as in **e** but imaged with 40 ms exposure time. The whole section is imaged with 120 ms exposure time **a**–**e**. **g** SNRs of the images (**e, f**) are computed with the same procedure as Fig. 2d. Scale bars, 100 μm **a**, 10 μm **b**, **c**, 2 μm **d**, 1 μm (**e**, upper), 200 nm (**e**, lower).

montage imaging (Supplementary Fig. 1e, Supplementary Fig. 4a, b). Over 3000 sections of hippocampus CA3 tissue, each 1 mm² in size, have been imaged with bdTEMs at a pixel size of 3 nm. Overlap between supertiles is 10% and overlap between subtiles within a supertile is 15% (Fig. 3a–e, see Data Availability). No additional pre-irradiation of the imaging region is needed; however, imaging the central first tile typically would expose neighboring tiles to electron beam irradiation and cause section deformation. All sections are stitched with AlignTK, which utilizes an elastic spring mesh algorithm capable of handling non-linear distortion owing to lens or electron irradiation. AlignTK has been used successfully for multiple TEM datasets[26,27,29,47]. Ultrastructural features such as postsynaptic densities and vesicles can be clearly defined (Fig. 3e, f). Small-diameter axons and dendrites can be followed across sections. Two cutout volumes, one imaged at 120 ms exposure time and the other at 40 ms, are aligned and segmented (Fig. 4, Supplementary movie 1-2), using our deep learning-based reconstruction pipeline[48]. Qualitative comparison from these cutouts shows the segmentations from two imaging conditions are similar (Fig. 4d). This is not surprising, given that down-sampled images at 8 or 16 nm per pixel have been shown to be adequate for high-quality alignment and segmentation[48,49]. These data

demonstrate that sequential sections imaged with the bdTEMs can be assembled into a 3D volume to reconstruct neural morphology and connectivity.

## TEM facility
An array of four microscopes have been fully installed in our facility. In general, 1–2 microscopes are offline for repairs or allocated for testing and development, while the rest have accumulated many months of consistent imaging time with the exception of filament replacements and routine maintenance. By estimate, if the four bdTEMs are running 24/7, imaging a cubic millimeter will take about a month (Table 1). However, non-stop long-term imaging requires further development of automation software that can handle image quality control, work-flow management, image and section databases, and stitching and alignment. A scalable architecture with these functionalities has been developed[31] but needs to be adapted to integrate with bdTEMs.

## Discussion
Here we report a substantial increase in TEM imaging speed enabled by beam deflection technology. Beside its technical benefit, this technology is more accessible. Similar to previous effort to automate

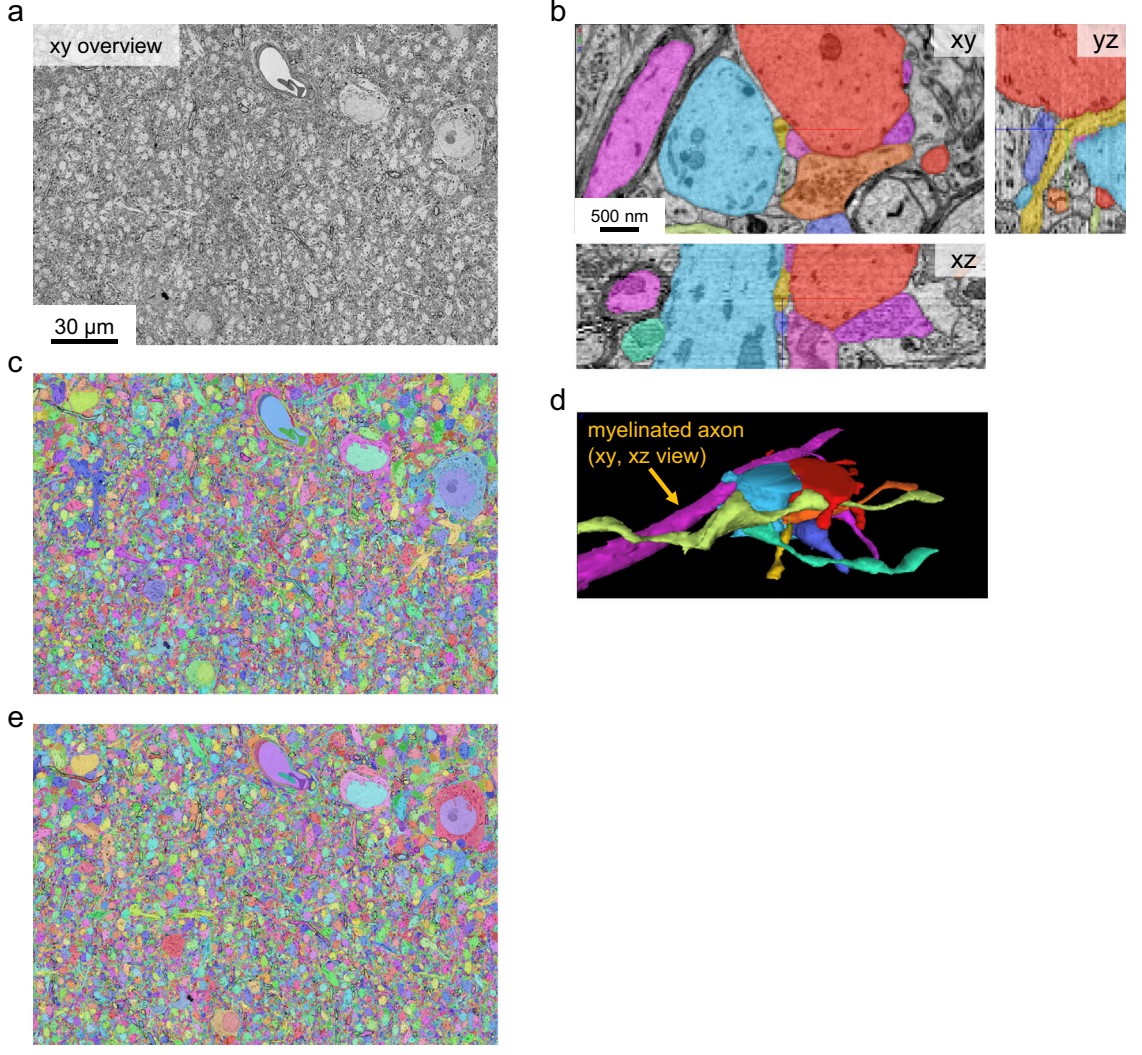

**Fig. 4 | Segmentation of bdTEM imaged data.** Alignment and dense segmentations of two cutout volumes of the same area in hippocampus CA3 region imaged with bdTEM. Results of 1 independent experiment. **a–d** EM overview and segmentation of a cutout volume imaged with 120 ms exposure time ($191 \times 151 \times 2.48\ \mu m^3$, 55 sections); **e** Segmentation of a cutout volume imaged with 40 ms exposure time ($191 \times 151 \times 0.9\ \mu m^3$, 20 sections).

TEM[28,29], our modifications are based on a retrofitted JEOL 1200EX-II microscope and cost for the entire system (<$500,000) is significantly lower compared to equivalent imaging systems, permitting acquisition of multiple TEMs for an imaging facility (six at the Allen Institute for Brain Science[28] and four at Princeton University). The space requirements for such an imaging array is modest given its productivity, spanning an area of 9.3 meters by 5.6 meters in total for 4 microscopes. The current system, however, requires specialized films on GridTape, which scales linearly in cost with the number of sections (USD $4 - $8 per slot); but improving film-making methods and economies of scale may decrease the cost in the future. While some custom engineering, such as automation of serial sectioning and sample delivery, is needed to implement a connectomics imaging pipeline, beam deflection by itself only makes use of the existing beam shift, image shift deflectors without needing to re-engineer microscope optics. These deflectors are standard components of TEMs from any manufacturer.

TEM has the advantages of high spatial resolution up to atomic scale and better imaging depth, but suffers from limitation in field of view size. Imaging a large area is classically slow because a large number of stage movements are required. Beside connectomics, our approach of combining beam deflection and fast stage movement can be used to expand imaging area and increase imaging speed in many other TEM applications, including cell biology[50], clinical pathology[51], and material science[52].

The examinations of imaging overheads (Fig. 1e) and beam deflection limits (Fig. 2a) points to several avenues of future improvements in raw imaging speed. (1) Computational overhead can be reduced by refactoring frequently used tasks in the acquisition software to low-level programs. (2) The current 15% subtile overlap within a supertile can be further reduced, for example, to 8%[28]. (3) Upgrading the optics with a 100 MPix camera will in principle double the burst imaging rate. (4) We have initially used 3 by 3 subtiles for large-scale imaging since one circumference of subtiles minimally demonstrates the benefits of beam deflection. Adding more subtiles should further increase overall imaging speed but return would be diminished given the proportion of stage movement overhead is already down to 4% of total time, with trade-offs of lower SNRs for the tiles located at the peripheries. (5) Given that downsampled data (for instance, at a pixel size of 8 or 16 nm) have proved to be adequate for high-quality alignment and segmentation[48,49], one potential approach to increase imaging speed per unit area of tissue is to either decrease magnification (thus increasing pixel size) or further decrease exposure time, with the caveat that the loss in information content be carefully evaluated against groundtruth. In fact, fine alignments of volume

**Table 1 | Performance metric for TEM with Cricket**

| imaging parameters | avg. 3 nm/pix (120 ms exp.) | avg. 3 nm/pix (40 ms exp.) | peak 3 nm/pix (40 ms exp.) | avg. 3.6 nm/pix (40 ms exp.) | unit |
|---|---|---|---|---|---|
| x,y pixel size* | 3 | 3 | 3 | 3.6 | nm/pixel |
| tile size (per size) | 6000 | 6000 | 6000 | 6000 | pix |
| FOV size | 18 | 18 | 18 | 21.6 | μm |
| pixels per tile | 36 | 36 | 36 | 36 | Mpixel |
| supertile size (per side, 15% overlap b/w tiles) | 16200 | 16200 | 16200 | 16200 | pix |
| overlap between supertiles | 600 | 600 | 600 | 600 | pix |
| exposure time (exp.) | 120 | 40 | 40 | 40 | ms |
| tiles per section (w/overlap) | 4356 | 4356 | 4140 | 2916 | tile |
| supertiles per section (w/overlap) | 484 | 484 | 460 | 324 | supertile |
| imaging time per section | 11.7 | 6.6 | 4.7 | 4 | minute |
| transition overhead (tape translation, ROI definition, autofocus, etc.) | 2.2 | 2.2 | 1.8 | 2.2 | minute |
| total time per section | 13.9 | 8.8 | 6.5 | 6.2 | minute |
| burst imaging rate (imaging only) | 300 | 900 | 900 | 900 | Mpix/s |
| montage imaging rate (w/ stage, Cricket) | 223 | 397 | 529 | 437 | Mpix/s |
| net imaging rate (w/ stage, Cricket, transition) | 188 | 297 | 382 | 282 | Mpix/s |
| effective imaging rate (w/ stage, Cricket, transition, overlap) | 142 | 224 | 288 | 213 | Mpix/s |
| sections per day (1 scope at 24 h) | 103 | 163 | 221 | 232 | section |
| time to image 1 mm³ (4 scopes at 24/7) | 54 | 35 | 26 | 24 | days |
| time to image 1 mm³ (4 scopes at 65% uptime #) | 83 | 53 | 39 | 37 | days |

Based on 1 mm² section of 45 nm thick.

# 65% uptime accounts for microscope maintenance, imaging pause (Yin et al. 2020).

*Pixel size is different from resolution, which refers to the capability of an imaging system to perceive two point sources separately.

Typical TEM section thickness is 45 nm in z, while FIB-SEM offers a z resolution of 4 nm (Xu et al. 2017).

cutouts show no difference between images acquired with 120 ms and 40 ms exposure time (Supplementary movie 1-2). Given that alignment quality is a major limiting factor influencing segmentation accuracy[48], imaging at 40 ms exposure time should be sufficient. In ongoing large-scale imaging efforts, we are routinely using exposure time of around 40 ms.

Volume EM is dichotomized into SEM-based and TEM-based methods, and both have been shown to be capable of acquiring petascale datasets (MultiSEM[17,53] and GridTape-based TEM[30]). Multiple parallel electron beam technology is based on sophisticated innovations of illumination and detection optics[12–14]. On the other hand, the improvements in TEM for large-scale connectomic applications are largely focused on stage mechanics, section delivery, and camera sensor[2,28,29]. The motivation is to automate or improve other components to leverage the inherent parallel imaging capability of the transmission mode. Crucially, both imaging methods currently depend on cutting and collecting a large number of ultrathin sections. Serial section EM imaging, including ssTEM in this paper, offers excellent lateral resolution (Table 1; Supplementary Fig. 2) but axial resolution is limited by section thickness, typically at 35−50 nm. Furthermore, sectioning-related artifacts (folds, cracks, knife marks, etc.) are very common, though automated reconstruction methods that are robust to some defects have been developed[49]. In this regard, ion beam milling combined with SEM imaging, not only circumvents ultrathin sectioning but also provides isotropic spatial resolution of 4 nm per pixel[10,11]. In particular, the recently developed wide-area milling of imaged tissue[54,55], when integrated with a MultiSEM, can potentially scale up connectomic volumes beyond petascale. In addition, the reliability of serial sectioning can be improved by magnetic section collection[56]. For TEM, one proposal to improve reliability of sectioning is to increase the thickness of serial sections, which are then imaged with tomography to recover a finer axial resolution[50,57]. Whether SEM or TEM-based techniques is the method of choice for exascale datasets remains to be determined.

## Methods

### Sample preparation and sectioning

All procedures were carried out in accordance with the Institutional Animal Care and Use Committee at Princeton University. The animal is housed in reverse light cycle conditions (light cycle: 8PM−8AM), temperature was 70 ± 2 °F, and humidity was 50 ± 10%. A female mouse (*Mus musculus*, C57BL/6J-Tg(Thy1-GCaMP6f) GP5.3Dkim/J, Jackson Laboratories, 028280) aged 4 months was transcardially perfused with a 4 °C fixative mixture of 2.5% paraformaldehyde and 1.3% glutaraldehyde in 0.1 M Cacodylate with 2 mM CaCl$_2$ pH 7.4. The brain was extracted and post-fixed for 36 h at 4 °C in the same fixative solution. The perfused brain was subsequently rinsed in 0.1 M Cacodylate with 2 mM CaCl$_2$ for 1 hr (3 × 20 min, 4 °C) and 300 μm coronal sections were cut on a Leica Vibratome on ice. Sample blocks were cut out and stained based on a modified reduced osmium (rOTO) protocol with the addition of formamide[30,58]. The tissue blocks were first *en bloc* stained with 8% formamide, 2% osmium tetroxide, 1.5% potassium ferrocyanide for 3 hours at room temperature. Subsequently, the samples were immersed in 1% thiocarbohydrazide (TCH, 94 mM) 50 °C for 50 mins, followed by a second step of 2% osmium staining for 3 hours at room temperature. The sample was placed in 1% uranyl acetate overnight at 4 °C, followed by lead aspartate (Walton's, 20 mM lead nitrate in 30 mM aspartate buffer, pH 5.5) at 50 °C for 2 h. After washed with water (3 ×10 mins, room temperature), samples proceeded through a graded acetonitrile dehydration series[59] (50%, 75%, 90% w/v in acetonitrile, 10 min each at 4 °C, then 4 ×10 min of 100% acetonitrile at room temperature). After a progressive resin infiltration series (33% resin:acetonitrile, 50% resin:acetonitrile, 66% resin:acetonitrile, 8 hours each at room temperature), the sample was incubated

in fresh 100% resin overnight at room temperature and the resin was cured in the oven at 60 °C for at least 48 hrs.

GridTape (Luxel Corporation) contains regularly spaced apertures with plastic film substrates for serial sections[28,29]. The films are transparent to electrons, and therefore are compatible with TEM imaging. We combined an automated tape collecting system (ATUMtome, RMC/Boeckeler) and an ultramicrotome (UC7, Leica) to create an automated tape-based sectioning and collection system (Supplementary Fig. 3) for GridTape, based on previous designs[28,29]. Our custom ultramicrotome setup also includes a computer-controlled, high-precision motorized stage, monitoring of temperature and humidity, and three cameras to monitor the collection process. Additionally, we have built closed-loop control software to phase-lock aperture movement with cutting, in order to center sections collected in the apertures on GridTape. After resin embedding, the ultrathin sections were then cut at a nominal thickness of 40 nm and automatically collected onto a GridTape.

### Reel-to-reel translation system and GridStage
Gridstage Reel[28] comprises three major components: a stage cartridge, a reel storage and delivery system, and an airlock assembly enabling cartridge loading. A 3D rendering of the GridStage Reel system is shown in Supplementary Fig. 1, attached to a JEOL 1200EX-II microscope chassis. GridStage's two-axis stage cartridge is set to a fixed height to enable the imaging environment to stay the same from section to section as they are delivered into the imaging area by conveyor. Each axis is driven by a precision closed-loop piezo motor. The precision is ~50 nm, and absolute positional accuracy on the sample is typically between 100 and 200 nm. This mechanism positions the sample accurately and reliably within the requirements of montage image overlap reproducibility, even given electron-optical environment variability (e.g. charging and sample morphology changes) of different samples in the volume. The cartridge incorporates an accurate barcode reading and clamping mechanism to precisely position and identify slots on GridTape, with read accuracy greater than 99.9% for a new reel of GridTape. This enables precise tracking of the current tissue section under first-time section imaging or during re-imaging operations. The precision of the GridStage sample ID subsystem supports either sequential or fully random-access sample delivery modes.

The stage has fast responsiveness, with roundtrip communication to the drive system on the order of a few milliseconds, and typical step-and-settle times occurring in under 45 ms (Supplementary Fig. 1c, d). The stage motion time is slightly improved over a previously reported fast Piezo-driven stage[2]. The distributions of stage motion time are spaced at intervals of 2 ms; this stage motion artifact likely exists during actual imaging. While the reason for this splitting has not been accounted for, this effect has minimal influence on the total imaging time given that the average time is relatively fast. During imaging, an extra 10 ms of settle time is added for each movement to ensure stability. The 10 ms per step was taken into account in the stage movement overhead.

Voxa reels are housed in robotic delivery and take-up conveyor systems connected to the microscope's vacuum, and can accept a GridTape reel with up to 7500 sections spaced 6 mm apart. A continuously-monitored belt tension meter ensures the GridTape translates within specified operating parameters, and supports safe transit of the samples into and out of the imaging area. The GridStage reel sample delivery system can deliver sequential sections as fast as once every two seconds, for quick survey modes.

### Lens assembly and camera
The electron optics of each of our TEM systems consists of a custom phosphor scintillator of 75 mm in diameter, a lens assembly (AMT, NanoSprint50M-AV), and a high speed CMOS camera (XIMEA, CB500MG-CM). The pickup area of $47 \times 47$ mm on the phosphor screen is near the film plane of the JEOL 1200EX-II TEM and therefore minimizes image distortion at the bottom of the column and matches the nominal magnification setting of the microscope platform. Conveniently, this setup eliminates the need for a lengthened column that previously requires building scaffolds and is impractical for many facilities[2,26,28]. In addition, compared to the previous design of an extended column, the smaller scintillator and compact lens in our TEMs allows good signal-to-noise with lower dose or with shorter exposure times leading to more efficient use of available beam electrons. The CMOS camera has a PCIe interface that supports fast data transfer rates. Overall, the lens and camera support nominal image acquisition with a frame size of $6000 \times 6000$ pixels at a shortest exposure time of 40 ms. In the future, an upgrade of a 100 Mpix CMOS camera with backside illumination could potentially lead to another couple-fold increase of imaging rate[28].

### Beam deflection
The beam deflection module (Cricket™, Voxa) for TEM has been prototyped previously[28] but not demonstrated for large-scale imaging. In bdTEM, the illumination of the sample in the TEM, which normally passes the beam through the sample on the optic axis, is rapidly and sequentially routed in a coordinated fashion to sample a pattern of locations adjacent to the optic axis. Each location fills sub-sections (subtiles) of a virtual composite camera image (supertile) centered on each stage position in the montage. At each beam-deflected subtile position (scan), the image formed there is precisely and symmetrically un-deflected (descanned) back to the optic axis to illuminate the TEM camera.

The scan/de-scan is achieved using the shift-tilt, image shift, and projector deflectors in the JEOL 1200EX II, driven by Voxa Cricket™ beam deflection hardware configured and calibrated in Voxa Blade™ software. To minimize the impact to image resolution due to lens aberrations, the scan is conducted by calibrating the deflected beam to precisely pass through the coma-free rocking point of the objective lens. In the de-scan, buildup of projection parallax distortion along with the projector distortion is avoided by precisely returning the beam to the optic axis.

The pattern of deflected positions is selected to rapidly and efficiently acquire subtiles for compositing into a supertile. The nominal minimal sampling arrangement is a $3 \times 3$ array of subtiles with center tile on the optic axis, which enables an increase in areal acquisition efficiency per stage move by about an order of magnitude. The flexible system design enables myriad subtile configurations, for example hexagonal close packing, cross patterns, aliased circular patches (e.g., 2-4-2), or line patterns.

### TEM imaging
Some components of the TEM systems have been briefly presented before[60]. For imaging, samples on a GridTape reel are first loaded into the reel housing of the reel-to-reel system (Supplementary Fig. 1a), which is connected with the TEM column vacuum. After loading, the microscope is pumped down to reach the vacuum level of ~1E-7 Torr. Lanthanum hexaboride crystal ($LaB_6$) filaments were used due to their high electron flux per unit area and a longer lifetime (1000 h or more as opposed to 200 h for a tungsten source) reducing downtime needed for filament change. Crucially $LaB_6$ filaments depend more sensitively on a good vacuum pressure for good stability and lifetime (low $10^{-7}$ Torr). It typically takes about half a day when loading a new sample reel to pump down the column to the base operating pressure for LaB6. We then increase the high-tension voltage to 120 KV, turn on filament current and then perform alignments on various components of the TEM, following routine technical instructions for TEM operations. These procedures take half a day and only need to be performed after installation of a new filament or loading of a new tape. Calibration of reel-to-reel systems involves tape and tension calibration for reliable translation and barcode reading, which typically takes ~5 min and is only done when first loading a reel. Cricket alignment is performed to ensure each sub-tile has sufficient overlap (~15%) with neighboring sub-

tiles and that there is minimal distortion across the field between images from different sub-tiles to facilitate efficient stitching and reconstruction. Cricket alignment is usually stable over several months. The TEMs are compatible with GridTape[29], which has regularly spaced $2 \times 1.5$ mm$^2$ apertures in aluminum-coated polyimide tape. Each aperture can be identified by a unique barcode milled on the tape. The tape is coated with a 50 nm-thick film (Luxel Corporation) that spans the apertures and serves as support for sections. In a recent large-scale imaging effort, 55 sections need to be re-imaged for a reel of 1785 sections. The re-imaging rate is therefore 3%.

The acquisition computer for each microscope has an intermediate storage of 16–32 TB SSDs via a PCIe card with 4 USB3.0 ports, for ~24 h of imaged data. Each USB port can support up to 10 Gb/s of transfer speed. After caching on the temporary storage on SSD, the data are then transferred via a 10 Gb/s network connection to an on-premise petascale cloud storage in a separate building. Data transfer is done using CloudFiles (https://github.com/seung-lab/cloud-files).

### Acquisition software

The steps to perform before imaging every section include: deliver a new aperture into the field of view using the reel-to-reel system, locate the aperture center using stage algorithmically, find the illumination center of the electron beam (which is often off-center due to magnetic hysteresis from magnification change or change in charge equilibrium state of the new aperture), adjust camera gain control, extract ROI, and autofocus at high magnification (Supplementary Fig. 4-5). These are done for every aperture with a tissue section, and typically take 2 – 3 minutes per aperture (Table 1). Software (Blade, Voxa Inc.) has been developed that can control the Reel-to-reel system, GridStage, and TEM automatically. The software has lower-level functions that execute each step in the workflow (e.g. travel to a specific aperture in the GridTape reel, move the stage, adjust focus of the TEM). The low-level software supports end-user scripting and control of application-specific microscope workflows via the Blade API supplied to the user. Additionally, the Blade software allows initialization of imaging remote control over IP of all operations and multiple user-defined imaging modes - e.g. continuous batch imaging, re-imaging, and selective or random sampling.

### Quantification of signal-to-noise power ratios (SNR) and distortion

We used a previously described method[45,61] to determine SNR as a function of spatial frequency for images. Two-dimensional spatial power spectra were computed for each image and collapsed into one dimension by radial averaging. Electron beam was blocked to collect noise images and the power spectra of noise images were computed using the same procedure and subtracted. One contributing factor to the frequency domain measurement of SNR is focus quality, because sharper edges lead to increased magnitudes of high-frequency components[62]. Notably, fast Fourier transform-based algorithm was often used as a measurement of focus quality in large-scale imaging efforts[28]. The magnitude of SNRs are different between Fig. 2d and Fig. 3g, because different magnifications were used (3 vs. 3.6 nm/pix, respectively) and, as a result, the electron doses are different.

Non-rigid distortion was quantified by assessing the displacement of image features before and after stitching. Local image features were automatically extracted from pre-stitching images in an evenly spaced manner. The corresponding locations of these features in post-stitching images were determined through block matching using normalized cross-correlation[63].

### Stitching, alignment, and segmentation

Image stitching was performed with the AlignTK software[26,27] (https://mmbios.pitt.edu/aligntk-home). Image alignment and segmentation was performed with a custom software pipeline[48].

### Reporting summary

Further information on research design is available in the Nature Portfolio Reporting Summary linked to this article.

## Data availability

A dataset of fully stitched EM data is available to view interactively at https://seung-lab.github.io/CA3Montaged (z = 1013−1112). Image data are available for download at precomputed://https://s3-hpcrc.rc.princeton.edu/ca3-aligned-2023/image_cutout using the CloudVolume software (https://github.com/seung-lab/cloud-volume). Other data presented in the study are available from the authors on request.

## Code availability

Code is available from https://github.com/seung-lab/PyAGTUM, https://doi.org/10.5281/zenodo.11541305.

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

## Acknowledgements

The Cricket system was initially prototyped and tested by Voxa with the Allen Institute for Brain Science[28]. We thank R. Clay Reid, Daniel J. Bumbarger, Wenjing Yin, Derrick Brittain, Marc Takeno, Nuno Macarico da Costa, David Hildebrand for advice on microscopy and serial sectioning setup; Stephan Y. Thiberge for help in the planning and

construction of the TEM facility; Lawrence Own and Teddy DeRego for their development and support in GridStage and Cricket hardware and software; Wei-Chung Allen Lee and Jasper Phelps for advice on image stitching; John Price for modification of ATUMtome. This work was supported by the NIH grants 1S10OD023602-01A1 (H.S.S.), 1U19NS104648 (D.W.T., H.S.S.), 1U19NS132720 (D.W.T., H.S.S.), 1UM1NS132253-01 (H.S.S.), 1K99NS135650-01 (Z.Z.), and the Simons Collaboration on the Global Brain (D.W.T.). The authors acknowledge the use of Princeton's Imaging and Analysis Center, which is partially supported through the Princeton Center for Complex Materials (PCCM), a National Science Foundation (NSF)-MRSEC program (DMR-2011750).

## Author contributions

C.S.O., A.A.W. and H.S.S. conceived the study. Z.Z. developed the conceptual framework for quantifying the performance. D.W.T. and H.S.S. acquired funding and supervised. A.A.W. planned the construction of the facility and managed the installation and testing of the first instrument. A.A.W. registered and stitched the initial datasets. Z.Z., C.S.O., and R.A.K. developed software for TEM imaging. Z.Z., A.A.W., and E.W.H. developed software and hardware for serial sectioning. W.M.S. developed software for data transfer. Z.Z. performed image stitching. Z.Z. and N.K. performed image alignment. R.L. performed segmentation. Z.Z. acquired and analyzed data. Z.Z. and C.S.O. carried out validation and evaluation. Z.Z. and H.S.S. drafted the manuscript with input from all authors.

## Competing interests

N.K., R.L., and H.S.S. disclose financial interests in Zetta AI LLC. C.S.O. and R.A.K. disclose financial interests in Voxa. A.A.W. is a founder and owner of ariadne.ai ag (Switzerland). The remaining authors declare no competing interests.
