## [Peer Review File · Nature Communications]

REVIEWER COMMENTS

Reviewer #1 (Remarks to the Author):

In their manuscript entitled “Fast imaging of millimeter-scale areas with beam deflection transmission electron microscopy the authors describe an increased throughput in serial-section TEM imaging when applying a hybrid montaging approach that employs beam/image shift in addition to stage movements. While the speed-up of the experiment is convincing and the absolute imaging rate the system achieves is fascinating, the novelty described in the paper in my view is not sufficient to justify publication in a journal whose target audience is a broad scientific community. I encourage the authors to focus more on the technical aspects of their development and target a specific journal covering methods in ultrastructural imaging or perhaps even an engineering journal.

Major issues:

As stated, the amount of novelty presented in the manuscript is limited to the beam/image deflection mechanism in the microscope and associated software control. This development is described in merely 9 lines of the main text with another seven lines describing the increased throughput. All other aspects of the experiment, from specimen preparation to data handling and processing are already published previously or even commercially available. In addition to applying beam/image shifts, using a different type of camera and mounting could also be considered a novelty. Given these are the key statements of the manuscript, I would have hoped for more technical details of the developments. Providing a technical description of solely these developments would make such a manuscript a good candidate for a more specific journal.

The conceptual idea of combining stage and image shifts for “hybrid montaging” is well established and part of several established acquisition software solutions for microscopes and detectors of various manufacturers. The necessary deflector coils are part of a standard TEM. What makes the described approach different, is the speed and accuracy of the applied shifts, which indeed is remarkable. Unfortunately, the technical details on how this is achieved are missing in the paper.

Another major obstacle of applying or implementing the described developments in a broader scientific context is the complexity of installing and running such a high-throughput multi-microscope installation. Several technical and engineering staff is necessary to install and maintain the ensemble of custom hard and software. As described in the manuscript, the approach is currently used at two institutes focusing on and world-leading in whole-brain connectomics imaging. While such developments at the forefront of scientific discovery pushing the boundaries of how data can be collected and processed typically lead to a trickle-down effect, which also other

scientific fields can benefit from, I hardly envision this effect from the particular technical development described here (maybe due to the lack of detail).

Minor issues:

For better understanding of the scales, please use SI units throughout the manuscript.

Reviewers #2 and #3 (Remarks to the Author):

Faster imaging with beam deflection TEM

Major: The imaging potential is impressive, but...

The paper is poorly structured, with a non-compelling abstract (starting with: We have achieved...) and an introduction that is centered on some preceding work related to the microscope, but other initiatives that may result in similar achievements are not introduced and incompletely discussed. Regarding the introduction: Recent reviews (2022) on vEM are available that may be used to improve the description of the current state of the field, including the benefits (and disadvantages) of other methods: FIB-SEM, ssTEM, multi-beam SEM; multi-beam TEM etc.

The beam deflection maybe a new avenue for faster EM, a proof-of-concept (though with limited data) seems to be reached, but the pipeline is not novel and far from complete as highlighted by the authors, including the possible solutions/ expected improvements. (Actually, the manuscript reads more like a project proposal rather than a problem-solving assay). Only 'innovation' seems is the electromagnetic lens deflectors (Cricket) to kick the beam off-axis for "supertiles", which actually also was previously described (reference 12). The Cricket system is quite poorly described and characterized. There is a reference to ref 12, but actually there is not much more detail in ref 12 than in the current manuscript.

Detail is lacking on how Cricket is implemented in the electron optics path of the TEM, the Fig 1a is quite poor in content. What are the deflection angles? Is there any influence on image quality and can that be quantified for the different supertiles? In a focused beam system (SEM), beam deflection may give rise to focus blur and beam position errors, as examined in detail in Zhang et al, Ultramicroscopy 211 (2020) 112925 and other references therein. Everything else is improvements from their previous work. (Yin 2020).

Proper data is lacking - Yes, faster EM would benefit many biomedical projects, and yes, there is a data avalanche that may be a next bottleneck to analyse. However, data is hardly presented (snapshots), and should be made available to the community to (1) help to define the quality of images by this modality (highest resolution, all areas, stitching etc); (2) help to handle the challenges on such datasets (interpretation, sharing, metadata co-publishing).

Figure 2 (floorplan, pictures, including a WC for every operator) and accompanying legend and main text is redundant and does not add anything.

Performance analysis is concise, yet thorough. E.g. both average performance and peak performance specifications are included in table 1.

Table 1: how can 3nm/ pixel lead to 3nm resolution? Please emphasize that 'the resolution' is limited by the sample thickness, and maybe discuss that would be tackled better by using FIB-SEM.

Highly specific comments

56: would have expected some discussion on why 3x3 and what the limit here is.

68: not clear to me if whole dataset in Fig. 1 was acquired with 40ms or 120ms exposure time (40ms is sort of implied but not explicit).

68: insufficient resolution in Fig. 1i to determine difference in noise level / apparent SNR. 4x further zoom would be better. A quantification here would also be nice.

85: The 50Mpx camera sounds like a hypothetical comparison to the 36Mpx camera they are using (line 66), which is fine, but then extended data Fig. 1A has 50Mpx. Similar to confusion over where it is 40ms vs 120ms exposure time.

100: (spelling) subtitles should be subtiles.

108-109: what is "reasonable" microscope downtime?

117: Limits to reducing tile overlap?

117: Why enlarge the pixel size?

283: code and a paper about their software pipeline is linked, but data is not available?

Extended Data Fig. 1A: Nothing objectively wrong with it, but (b), (c), and (d) are not very pretty, messy organization and plots are bland.

Imaging should be imaging (Table 1)

Reviewer #4 (Remarks to the Author):

The authors report the real implementation of a beam deflection mechanism for increased acquisition speed of serial section electron microscopy data. This technique has been proposed in a previous study but was not implemented. Therefore, this work provides concrete evidence on the effectiveness of the bdTEM (beam deflection Transmission Electron Microscope) for rapid imaging of ultrathin serial section using TEM, and demonstrates that it is also possible to create a 3D reconstruction of the acquired images for detailed analysis of the connectomics.

The basic concept is well presented, but a broader reader would benefit by providing related introductory information, and some results need clarification. Points to be addressed are detailed below.

Major comments:

In the abstract the authors provided only information on the implementation itself, but the paper would benefit with a brief background information and additional discussion following the “guide to authors” of the journal.

In line 60, it states that stage movements accounts for only 7%, and that cricket settling time accounts for 9%, pointing to the graph in Fig. 1d. These values were not obtained from a single measurement, and therefore, statistical data should also be provided. Moreover, the definition of “tile overlap” is described in the “methods” section, but should also be present in the figure legend. Lastly, the duty cycle value of 30% is detailed in the “Methods” section, but the values should be presented in the main text, or a reference to the methods could be added. For this paragraph, a note referencing the “Methods” section could be added to guide reader to look at the “Methods” section to understand how each value was obtained.

In line 70 the authors points to Fig. 1e-i, but it should point only to Fig. 1i, as the text reports about imaging quality with 40 ms exposure time. In addition, when we look at the figure, the scale is not well visible and it lacks the description to which figure points to exposure time of 120 ms and which to 40 ms in the figure, although this information is present on the figure legend. Lastly, the reconstructed images of the Fig. seems to be from a 120 ms exposure time condition, but the text states that it is was acquired at 40 ms exposure time. This is confusing, and need clarification. If Fig. 1 shows montage at 120 ms, it should say so in the text, as well as in the legend, and add another figure for the case of 40 ms exposure time, or a comment pointing to the respective

Supplementary Video. Moreover, related to Fig. 1e-i, Fig. 1h contains an extra letter 'h' just above the scale bar.

In line 78, the authors claim that the step-and-settle time is ~57 ms on both axis. Extended Data Fig. 1d shows a double peak distribution. How would the authors explain the double peak, or the low appearance of time between 56.5 and 56.6 ms for the y axis? Is this a measurement artifact? Or is it a defect on the piezo drive? And why is the count for the x axis much larger than the count for the y axis? Shouldn't the total count be the same? Moreover, this value could be also represented in statistical format.

In line 82 the authors report that the acquisition time of a 1 mm² section took 8.6 min. This is the average of how many sections? What are the statistical values?

In line 83 the authors points to table 1. In table 1, the burst imaging rate for the 120 ms exposure time is incorrect. It reads 900 (which is in the case of 40 ms exposure time), and should probably read 300. The other values may be correct, but I suggest revising the whole table.

Minor comments:

In line 55, "above the object plane and below" is unnecessary, as they are described just after this sentence.

In line 58, "eliminating 8 out of 9 stage movements" could be considered as "plagiarism" as the exact same sentence appears in reference 12. Different form to express this should be used.

In line 77, the data provided is for the previous system, therefore, it should say "ranged", and not "range".

In line 145 and 147, 'CaCl₂' should be written 'CaCl₂' with a subscript '2'

Many steps in the 'Sample preparation and sectioning' subsection lacks temperature information.

In line 151 TCH first appears and have not been defined. Thiocarbohydrazide?

Additional comment:

For this paper to become a reference for bdTEM, and since it is compared to the Allen system which incorporates quality control mechanisms, information whether re-acquisition of images were necessary for the reconstructed 3D block could be discussed. Moreover, a comparison and discussion on the overall outcome between imaging and 3D reconstruction for 120 ms exposure time and 40 ms exposure time would also provide additional information on possible configurations to consider when building a similar system. For example, did the lower quality of the 40 ms acquisition required additional processing for stitching and alignment compared to the 120 ms condition? Have the authors addressed how the two configurations would react to automatic segmentation? Or could the authors provide any thoughts on that matter?

We thank the reviewers for the constructive comments, which we believe have helped greatly improve the manuscript. Based on reviewers' comments, we have now clarified the novelty of the paper. The beam deflection technology is a new TEM engineering product that was briefly proposed (Yin et al. 2020), but its performance, consistency, and limitations were not clear at the time. The goal of the current study is to provide a detailed characterization of this innovative technology. To address the reviewers' concerns, we have made a number of major changes. These changes include adding a schematic of beam deflection, quantifying imaging speed across hundreds of sections, testing the limit for number of subtiles acquired with beam deflection, quantifying signal-to-noise ratios across subtiles, adding high resolution supertiles, releasing 100 fully montaged sections imaged with beam deflection TEMs, and alignment and segmentation of two cut-out volumes. The results demonstrate many advantages: the increase in imaging speed is substantial; image quality is consistent in a reasonably large dataset; it works across a wide range of resolutions; additional subtiles can be utilized to further expand beam deflection imaging area. In addition, the system makes use of the existing deflectors in 1980s TEMs without re-engineering microscope optics, and is therefore cost effective. We believe beam deflection will benefit a broad audience of EM users in general and our characterizations are important in facilitating adoption of the technique.

REVIEWER COMMENTS

Reviewer #1 (Remarks to the Author):

In their manuscript entitled "Fast imaging of millimeter-scale areas with beam deflection transmission electron microscopy the authors describe an increased throughput in serial-section TEM imaging when applying a hybrid montaging approach that employs beam/image shift in addition to stage movements. While the speed-up of the experiment is convincing and the absolute imaging rate the system achieves is fascinating, the novelty described in the paper in my view is not sufficient to justify publication in a journal whose target audience is a broad scientific community. I encourage the authors to focus more on the technical aspects of their development and target a specific journal covering methods in ultrastructural imaging or perhaps even an engineering journal.

We have now reframed the paper to focus on presenting practical performance data, rather than technical development. As stated in the overall response, we believe beam deflection is not only technically impressive but also affordable, which should facilitate its adoption. The study will be of interest to the volume EM (Peddie et al. 2022), connectomics (Abbott et al. 2020; Kornfeld & Denk 2018), and cryo-EM communities, all of which have experienced rapid growth in recent years. The deflectors are standard components of TEMs from any manufacturer. Therefore, many other TEM applications can potentially benefit from beam deflection to expand imaging area or increase imaging speed, including cell biology (Redemann et al. 2017), microbiology (Yoshida et al. 2020), and clinical pathology (Erlandson 2009). These texts and references have been added to the Introduction (lines 27 - 37) and Discussion (lines 222 - 230). Overall, we believe the benefits and potential applications justify publication in a journal with a broad audience.

Major issues:

As stated, the amount of novelty presented in the manuscript is limited to the beam/image deflection mechanism in the microscope and associated software control. This development is described in merely 9 lines of the main text with another seven lines describing the increased throughput. All other aspects of the experiment, from specimen preparation to data handling and processing are already published previously or even commercially available. In addition to applying beam/image shifts, using a different type of camera and mounting could also be considered a novelty. Given these are the key statements of the manuscript, I would have hoped for more technical details of the developments. Providing a technical description of solely these developments would make such a manuscript a good candidate for a more specific journal.

The conceptual idea of combining stage and image shifts for "hybrid montaging" is well established and part of several established acquisition software solutions for microscopes and detectors of various manufacturers. The necessary deflector coils are part of a standard TEM. What makes the described approach different, is the speed and accuracy of the applied shifts, which indeed is remarkable. Unfortunately, the technical details on how this is achieved are missing in the paper.

While we have reframed the novelty of the paper, we agreed with the reviewer that additional details on beam deflection will improve clarity. An optics diagram has now been added (Fig. 1a). We have also added additional beam deflection data, including maximal number of tiles (Fig. 2a), quantification of SNRs (Fig. 2d), and imaging across a range of magnifications (Supplementary Fig. 2).

We thank the reviewer for pointing out the technical highlights of beam deflection, which is reflected in the performance metrics (Table 1) and quality of imaged dataset (Fig. 3 and Data Availability). The reference to existing software that combines stage movement and image shift (Leginon, Suloway et al. 2005; SerialEM, Mastronarde 2005) have been added (Line 77).

Another major obstacle of applying or implementing the described developments in a broader scientific context is the complexity of installing and running such a high-throughput multi-microscope installation. Several technical and engineering staff is necessary to install and maintain the ensemble of custom hard and software. As described in the manuscript, the approach is currently used at two institutes focusing on and world-leading in whole-brain connectomics imaging. While such developments at the forefront of scientific discovery pushing the boundaries of how data can be collected and processed typically lead to a trickle-down effect, which also other scientific fields can benefit from, I hardly envision this effect from the particular technical development described here (maybe due to the lack of detail).

We would like to point out that all components, including the beam deflection unit, reel-to-reel systems, camera, and lens assembly are commercially available (See Methods). In particular, the beam deflection makes use of the existing deflectors in TEMs without re-engineering microscope optics, and can be readily installed. However, the performance of this technology was unclear before the current study. Our study should facilitate dissemination of the technology. In addition to Allen and Princeton, Wei-Chung Lee's lab has achieved remarkable success with GridTape-based TEMs (Phelps et al. 2021 Cell; Kuan et al. 2022 bioRxiv; Nguyen et al. 2022 Nature). The technology is expanding as more labs are in the process of installing GridTape-based TEM imaging systems, per personal communications with Albert Cardona (LMB-MRC, UK), Yoshi Kubota (NIPS, Japan), Aaron Kuan (Yale, US).

Minor issues:

For better understanding of the scales, please use SI units throughout the manuscript.

The dimensions in now Supplementary Fig. 3a have been converted to SI units.

Reviewers #2 and #3 (Remarks to the Author):

Faster imaging with beam deflection TEM

Major: The imaging potential is impressive, but...

The paper is poorly structured, with a non-compelling abstract (starting with: We have achieved...) and an introduction that is centered on some preceding work related to the microscope, but other initiatives that may result in similar achievements are not introduced and incompletely discussed. Regarding the introduction: Recent reviews (2022) on vEM are available that may be used to improve the description of the current state of the field, including the benefits (and disadvantages) of other methods: FIB-SEM, ssTEM, multi-beam SEM; multi-beam TEM etc.

We have revised the abstract to provide context to the paper. We have also added reference to the vEM review (Peddie et al. 2022) and introduction of other SEM and TEM methods, including FIB-SEM (Heymann et al. 2006; Knott et al. 2008; Xu et al. 2017), multi-beam SEM (Eberle et al. 2015) and multi-beam TEM (Mohammadi-Gheidari et al. 2010; Ren and Kruit 2016). The text and references have been added to the first paragraph of Introduction.

The beam deflection maybe a new avenue for faster EM, a proof-of-concept (though with limited data) seems to be reached, but the pipeline is not novel and far from complete as highlighted by the authors, including the possible solutions/ expected improvements. (Actually, the manuscript reads more like a project proposal rather than a problem-solving assay). Only 'innovation' seems is the electromagnetic lens deflectors (Cricket) to kick the beam off-axis for "supertiles", which actually also was previously described (reference 12). The Cricket system is quite poorly described and characterized. There is a reference to ref 12, but actually there is not much more detail in ref 12 then in the current manuscript. Detail is lacking on how Cricket is implemented in the electron optics path of the TEM, the Fig 1a is quite poor in content. What are the deflection angles? Is there any influence on image quality and can that be quantified for the different supertiles? In a focused beam system (SEM), beam deflection may give rise to focus blur and beam position errors, as examined in detail in Zhang et al, Ultramicroscopy 211 (2020) 112925 and other references therein. Everything else is improvements from their previous work. (Yin 2020).

As stated in the overall response, we have clarified the goal of the study to be characterizations of the beam deflection technology. References have been added including Zhang et al. (2020) and others that have examined the technical limitations of beam deflection in detail. Beside the original Fig. 1a (now Fig. 1b), we have now added a TEM optic diagram (Fig. 1a) to indicate the deflectors that were used. We have now shown image quality of different subtiles (Fig. 2c) and added quantifications of signal-to-noise ratio (SNR) across subtiles (Fig. 2d).

Proper data is lacking - Yes, faster EM would benefit many biomedical projects, and yes, there is a data avalanche that may be a next bottleneck to analyse. However, data is hardly presented (snapshots), and should be made available to the community to (1) help to define the quality of images by this modality (highest resolution, all areas, stitching etc); (2) help to handle the challenges on such datasets (interpretation, sharing, metadata co-publishing).

We have now provided additional data of 100 sections that are imaged with beam deflection TEMs and fully stitched. Each section is 1 mm² in size (see Data Availability). The quality of stitching demonstrates consistency of the bdTEM imaging systems. We have also added data acquired with beam deflection at high resolution (Supplementary Fig. 2 and Lines 114 - 115).

Figure 2 (floorplan, pictures, including a WC for every operator) and accompanying legend and main text is redundant and does not add anything.

This figure is a supplemental figure and we believe it makes an important point that the space requirements for four microscopes is modest, supporting the affordability of the technology.

Performance analysis is concise, yet thorough. E.g. both average performance and peak performance specifications are included in table 1.

We thank the reviewer for the positive feedback.

Table 1: how can 3nm/ pixel lead to 3nm resolution? Please emphasize that 'the resolution' is limited by the sample thickness, and maybe discuss that would be tackled better by using FIB-SEM.

We have changed the term to 'x,y resolution' with a footnote to indicate that FIB-SEM offers a superior z resolution.

Highly specific comments

56: would have expected some discussion on why 3x3 and what the limit here is.

We thank the reviewer for stimulating us to consider this issue. We have tested the limit, and were pleasantly surprised that we could even acquire 30 (6x5) tiles with beam deflection. We have added data to show the limit of beam deflection (Fig. 2 and text). At the current time, the payoff of going beyond 3x3 is limited in our particular application because of other sources of imaging overhead. However, it could become advantageous in the future to go beyond 3x3, and this could also be useful in other applications. These discussions have been added (Lines 103 - 124 & lines 238 - 240).

68: not clear to me if whole dataset in Fig. 1 was acquired with 40ms or 120ms exposure time (40ms is sort of implied but not explicit).

The dataset was acquired with 120 ms exposure time. We have now made it clear in the main text (line 147) and in legends of figure 3.

68: insufficient resolution in Fig. 1i to determine difference in noise level / apparent SNR. 4x further zoom would be better. A quantification here would also be nice.

We have now included 4x further zoom in the figure (Fig. 3e-f) and a quantification of SNRs (Fig. 3g).

85: The 50Mpx camera sounds like a hypothetical comparison to the 36Mpx camera they are using (line 66), which is fine, but then extended data Fig. 1A has 50Mpx. Similar to confusion over where it is 40ms vs 120ms exposure time.

We are sorry for the confusion. The camera we use has a sensor size of 8,000 x 6,000 (48 Mpix) and was rounded by the manufacturer to market as a 50 Mpx camera. We are using the maximal inscribed square, which is 6,000 x 6,000 (36 Mpix). To avoid confusion, we have now removed the "50 Mpix" claim and instead described the camera as having an effective frame size of 36 megapixels throughout the manuscript.

100: (spelling) subtitles should be subtiles.

The typo has been corrected. Thanks.

108-109: what is "reasonable" microscope downtime?

By microscope downtime we are following the same terms used in a previous study (Yin et al. 2020). This footnote and reference has been added to Table 1.

117: Limits to reducing tile overlap?

117: Why enlarge the pixel size?

We have revised the sentences as follows to describe limits to reducing tile overlap and rationale and caveat for enlarging pixel size:

"The current 15% subtile overlap within a supertile can be further reduced, for example, to 8% (Yin et al. 2020)."

"Given that downsampled data (for instance, at a pixel size of 8 or 16 nm) have proved to be adequate for high-quality alignment and segmentation^{46,55}, one potential approach to increase imaging speed per unit area of tissue is to either increase pixel size (i.e. microscope magnification) or further decrease exposure time, with the caveat that the loss in information content be carefully evaluated against groundtruth."

283: code and a paper about their software pipeline is linked, but data is not available?

We have now added data for 100 sections that are stitched (see Data availability). The small volumes that are aligned for 120 ms and 40 ms exposure time are in supplemental video 1 and 2, respectively.

Extended Data Fig. 1A: Nothing objectively wrong with it, but (b), (c), and (d) are not very pretty, messy organization and plots are bland.

We have replaced and re-organized (b), (c), and (d) in Supplementary Fig. 1.

Imaging should be imaging (Table 1)

We thank the reviewer for pointing it out. This typo has been corrected.

Reviewer #4 (Remarks to the Author):

The authors report the real implementation of a beam deflection mechanism for increased acquisition speed of serial section electron microscopy data. This technique has been proposed in a previous study but was not implemented. Therefore, this work provides concrete evidence on the effectiveness of the bdTEM (beam deflection Transmission Electron Microscope) for rapid imaging of ultrathin serial section using TEM, and demonstrates that it is also possible to create a 3D reconstruction of the acquired images for detailed analysis of the connectomics.

The basic concept is well presented, but a broader reader would benefit by providing related introductory information, and some results need clarification. Points to be addressed are detailed below.

Major comments:

In the abstract the authors provided only information on the implementation itself, but the paper would benefit with a brief background information and additional discussion following the "guide to authors" of the journal.

We have now revised the Abstract, Introduction, Discussion by providing context and implication of our work.

In line 60, it states that stage movements accounts for only 7%, and that cricket settling time accounts for 9%, pointing to the graph in Fig. 1d. These values were not obtained from a single measurement, and therefore, statistical data should also be provided. Moreover, the definition of "tile overlap" is described in the "methods" section, but should also be present in the figure legend. Lastly, the duty cycle value of 30% is detailed in the "Methods" section, but the values should be presented in the main text, or a reference to the methods could be added. For this paragraph, a note referencing the "Methods" section could be added to guide reader to look at the "Methods" section to understand how each value was obtained.

We have added three paragraphs in Methods to describe in detail how stage movement and Cricket settling time proportions are calculated and how their variabilities are considered (lines 426 - 447). To address the statistics raised in this question and a subsequent question, we have collected section imaging rates for 472 sections (Fig. 1e, and legends of Fig. 1d, e).

We have added an explanation of "tile overlap" in the fig. 1d legend. The values of imaging duty cycle and a reference to the methods have also been added to the main text (line 99).

In line 70 the authors points to Fig. 1e-i, but it should point only to Fig. 1i, as the text reports about imaging quality with 40 ms exposure time. In addition, when we look at the figure, the scale is not well visible and it lacks the description to which figure points to exposure time of 120 ms and which to 40 ms in the figure, although this information is present on the figure legend. Lastly, the reconstructed images of the Fig. seems to be from a 120 ms exposure time condition, but the text states that it is was acquired at 40 ms exposure time. This is confusing, and need clarification. If Fig. 1 shows montage at 120 ms, it should say so in the text, as well as in the legend, and add another figure for the case of 40 ms exposure time, or a comment pointing to the respective Supplementary Video. Moreover, related to Fig. 1e-i, Fig. 1h contains an extra letter 'h' just above the scale bar.

For the current Fig. 3 (previously Fig. 1e-i), we have revised the text to only point to panels with 40 ms exposure time, made scale bars visible, added labels of 120 ms and 40 ms near the pictures, clarified the exposure time of each image in the legends, and added a comment pointing to the data at 40 ms exposure time (Supplementary Video 2). The extra letter 'h' has been removed.

In line 78, the authors claim that the step-and-settle time is ~ 57 ms on both axis. Extended Data Fig. 1d shows a double peak distribution. How would the authors explain the double peak, or the low appearance of time between 56.5 and 56.6 ms for the y axis? Is this a measurement artifact? Or is it a defect on the piezo drive? And why is the count for the x axis much larger than the count for the y axis? Shouldn't the total count be the same? Moreover, this value could be also represented in statistical format.

We have since improved software, which further reduced software latency of the stage motion control (x-axis mean 34 ms, y-axis mean 40 ms). We have collected time measurements of 2,500 steps for x and y axes, respectively. These measurements, along with the means and standard deviations, are added as Supplementary Fig. 1b-c. As pointed out by the reviewer, the distributions have peaks at intervals of 2 ms. We have added the following description to Methods:

"The distributions of stage motion time are spaced at intervals of 2 ms; this stage motion artifact likely exists during actual imaging. While the reason for this splitting has not been accounted for, this effect has minimal influence on the total imaging time given that the average time is relatively fast. During imaging, an extra 10 ms of settling time is added for each movement to ensure stability. The 10 ms per step was taken into account in the stage movement overhead."

In line 82 the authors report that the acquisition time of a 1 mm² section took 8.6 min. This is the average of how many sections? What are the statistical values?

As stated in the response to a previous question, we have collected section imaging rates for 472 sections (Fig. 1e). Based on these data, the mean acquisition time is now updated to 8.8 mins (s.d. 0.9 mins). A paragraph to explain how these values are obtained has been added in Methods (line 415 - 425).

In line 83 the authors points to table 1. In table 1, the burst imaging rate for the 120 ms exposure time is incorrect. It reads 900 (which is in the case of 40 ms exposure time), and should probably read 300. The other values may be correct, but I suggest revising the whole table.

We thank the reviewer for pointing out the mistake. In Table 1, burst imaging rate for 120 ms exposure time has been corrected to 300.

Minor comments:

In line 55, "above the object plane and below" is unnecessary, as they are described just after this sentence.

This phrase has been removed.

In line 58, "eliminating 8 out of 9 stage movements" could be considered as "plagiarism" as the exact same sentence appears in reference 12. Different form to express this should be used.

We thank the reviewer for pointing it out. The sentence has been re-written.

In line 77, the data provided is for the previous system, therefore, it should say "ranged", and not "range".

The word has been corrected.

In line 145 and 147, 'CaCl₂' should be written 'CaCl₂' with a subscript '2'
We have subscripted the '2'.

Many steps in the 'Sample preparation and sectioning' subsection lacks temperature information.

All temperature information for the sample preparation steps has now been added.

In line 151 TCH first appears and have not been defined. Thiocarbohydrazide?

The full name for TCH, thiocarbohydrazide, has been added.

Additional comment:

For this paper to become a reference for bdTEM, and since it is compared to the Allen system which incorporates quality control mechanisms, information whether re-acquisition of images were necessary for the reconstructed 3D block could be discussed. Moreover, a comparison and discussion on the overall outcome between imaging and 3D reconstruction for 120 ms exposure time and 40 ms exposure time would also provide additional information on possible configurations to consider when building a similar system. For example, did the lower quality of the 40 ms acquisition required additional processing for stitching and alignment compared to the 120 ms condition? Have the authors addressed how the two configurations would react to automatic segmentation? Or could the authors provide any thoughts on that matter?

We have performed stitching, alignment, and segmentations of two small cut-out volumes, acquired with 120 ms and 40 ms exposure time, respectively. The results are shown in Supplementary Video 1 and Fig. 4 c for 120 ms and Supplementary Video 2 and Fig. 4e for 40 ms. The texts describing these results have been added (lines 180 - 193).

REVIEWER COMMENTS

Reviewer #1 (Remarks to the Author):

The authors have made major improvements to their manuscript such that it fits the more general scope of audience and highlights the novel technique and its benefits in a more comprehensive way. In my opinion, an updated version of the manuscript would be suitable for publication with some additional changes as highlighted below.

general points:

- While mentioning various SEM techniques in discussion part, a quantitative comparison of (volume) imaging performance is lacking in the introduction/results. It would help the reader to not only have an idea about the gain in throughput compared to non-bd serial TEM but also know about the reported values for the SEM-based approaches used in connectomics volume-EM imaging.

- while the technical details of the detection optics modifications (l. 143) or stage (l. 126) are provided, the key technological novelty (which is the beam deflection) of this publication still lacks a detailed description (l. 90). Please elaborate on the modifications to the microscope (electronics hardware etc.) needed to install the Cricket system. Also, there is no detailed description of Cricket in the methods part (in contrast to stage and detection optics which were described already before). Please provide more detail information about the deflection scanning procedure here as well. The spiral arrangement is only illustrated in Fig. 1 but mentioned nowhere in the text. What is the advantage over other (zigzag,...) geometries? I suggest to also discuss the dedicated acquisition software and its features (automation and quality control) in more detail (such as a screen grab as supplementary figure).

- Please provide information on the section deformation happening upon electron irradiation. Thin sections shrink and deform when first illuminated by electrons. This is especially true when applying the high flux that LaB6 filaments provide. How does this affect the bd acquisition? Does the beam for the central first tile already illuminate all neighboring tiles as well? Do you pre-irradiate the region to prevent shrinking? Do you correct for these effects post acquisition using non-rigid transformations when stitching the tiles?

- The software repository of pyAGTUM does not provide any documentation or installation instructions. Using a Leica ultramicrotome with an ATUM collector is of wide interest for the volume-EM community. An accessible solution (including the software) will be highly appreciated.

details:

- data availability: Is the source link correct? Pasting it into the source field results in an error while the direct hyperlink works.

Reviewer #1 (Remarks on code availability):

There is no sufficient documentation available to install or use the software.

Reviewer #2 (Remarks to the Author):

The revised version did address many of the points raised previously, and the manuscript has been improved.

- However, the data is still not available to address at full resolution and only limited data is available. Can data be shared via IDR or BIA? Proper sharing is essential for review purposes as well as for the final readership of NC, if the paper will be accepted.

- Suppl. F4 is redundant

- Discuss resolution. Previously, this was incorrect (3nm pixelsize = 3 nm resolution). Now it is no longer quantitative discussed.

- I did not check all points, because access to the raw data at full resolution is crucial to give further comments

Reviewer #3 (Remarks to the Author):

The manuscript has clearly improved. Authors revised their manuscript to go beyond mere technical realization, but if main message is not on the technical realization, the purpose of this development is to generate useful data and thus to either have the data available to the community, or present further analyses based on that data. Insight into data and data quality has clearly improved, but I feel that a high-impact publication on a disruptive data generating pipeline should present that data in full to the community, and not a subset of sections.

The manuscript is now better embedded in the context of EM development, but I think relevance to cryo-EM and microbiology is far-fetched. VolumeEM incl connectomics and cellular biology and

potentially extending to pathology in future is a very active, highly relevant field, no need to 'oversell'.

Abstract is now concise and stresses novelty, but the last sentence should be more specific for a broad audience: an array of how many in how many weeks?

There is a discrepancy in the comparison to previous literature: line 57 states prior state of the art achieved cubic millimeter in 6 months, while line 97 states authors achieve a two-fold increase in duty cycle.

The discussion and analysis of deflection and quality of deflected fields has improved, but the ray diagram added in Fig 1a is confusing. It seems there is a cross-over at the image plane, suggesting focused probe and no overlap with the deflected field. I think a better representation of the transmission system should be given with a (virtual) source magnified onto the sample plane and then projected onto an area on the camera. It also makes me wonder whether it is realistic that the beam still expands so much after the sample plane.

Fig 2 is a great and needed addition to the paper. For Fig. 2d I think the y-axis is the spectral-SNR, with the SNR being the integral over the full spectral width. It would be nice to tabulate SNR's for the subtiles, or have this values in a panel 2e like in the inset in Fig 2d.

All specific comments have been addressed.

Reviewer #4 (Remarks to the Author):

The authors have adequately addressed the reviewer's comments, and overall, the manuscript is well-crafted. However, I found some comments at this occasion with careful reading the manuscript. There is a challenge in understanding the effectiveness of integrating Cricket into the overall system, as well as the efficiency of increasing the number of tiling cards by Cricket from 9 to more than 9.

The authors have included a validation of the number of tiling sheets in the cricket, ensuring it exceeds 9. Although the evaluation is based solely on SNR, it is anticipated that distortion and focus blur would also increase. Would it not be more advantageous to provide a quantitative assessment of these aspects as well?

Furthermore, when considering a cricket with 9 tiling sheets, is it not essential to assess distortion and focus blur? Particularly, should these not be evaluated at the periphery of the diagonal image? Additionally, there might be a need for distortion correction; therefore, it would be beneficial to elaborate on the methodology employed for such correction.

It would be beneficial for readers if the authors could provide a comparative illustration for the following scenarios in a format consistent with Fig. 1d:

Without Cricket

With Cricket using 9 tiles

With Cricket using more than 9 tiles (e.g., 16 crickets)

Overall response

We thank the reviewers for their insightful comments. We are pleased that overall the reviewers appreciate the changes made in our last revision. For this revision, main requests from the reviewers include releasing all data, improving documentation for open-source software, providing more technical details of the beam deflection mechanism. While we share the reviewer's enthusiasm for publicizing more data, we respectfully maintain that 100 sections at high resolution are sufficient to demonstrate the feasibility of a new imaging pipeline. Acquisition and processing of a full EM dataset requires time and efforts that are beyond the scope of this manuscript, and in the field was typically published as a major piece of work in itself. As for software documentation, the public pyAGTUM code repository has now been populated with an explanation of key functionality, a 20-page manual, and a set up guide with hardware part numbers. We have also improved the ray diagram illustrating beam deflection and included additional technical information of beam deflection in Methods. While we understand that more technical details could be included, the beam deflection unit is readily available as a product to facilitate its dissemination. Furthermore, according to the reviewers' requests, we have revised the main text, methods, and figures (Fig. 1a), while also including new figures (Fig. 2e; Supplementary Fig. 1b,f; Supplementary Fig. 4&5). Overall, we hope the now much improved manuscript is deemed satisfactory for acceptance.

Point-by-point response

Reviewer #1 (Remarks to the Author):

The authors have made major improvements to their manuscript such that it fits the more general scope of audience and highlights the novel technique and its benefits in a more comprehensive way. In my opinion, an updated version of the manuscript would be suitable for publication with some additional changes as highlighted below.

general points:

- While mentioning various SEM techniques in discussion part, a quantitative comparison of (volume) imaging performance is lacking in the introduction/results. It would help the reader to not only have an idea about the gain in throughput compared to non-bd serial TEM but also know about the reported values for the SEM-based approaches used in connectomics volume-EM imaging.

We thank the reviewer for helping us improve the introduction. Recently reported multibeam SEM imaging rates have now been included (lines 35 - 41).

- while the technical details of the detection optics modifications (l. 143) or stage (l. 126) are provided, the key technological novelty (which is the beam deflection) of this publication still lacks a detailed description (l. 90). Please elaborate on the modifications to the microscope (electronics hardware etc.) needed to install the Cricket system. Also, there is no detailed description of Cricket in the methods part (in contrast to stage and detection optics which were described already before). Please provide more detail information about the deflection scanning procedure here as well. The spiral arrangement is only illustrated in Fig. 1 but mentioned nowhere in the text. What is the advantage over other (zigzag,...) geometries? I suggest to also discuss the dedicated acquisition software and its features (automation and quality control) in more detail (such as a screen grab as supplementary figure).

We have now included in Methods a new section with three paragraphs to describe the beam deflection (lines 370 - 390). We have also enhanced the ray diagram with greater detail and clarity (Fig. 1a). We opted for a spiral sequence beginning from the center tile, because the first, non-deflected tile, has the best image quality (Supplementary Figure 1b). Two supplementary figures that illustrate the acquisition software and its features have now been included (Supplementary Figure 4, 5). Although we acknowledge that certain specifics regarding the modifications, such as electronics hardware, have been omitted, this should not affect the dissemination of the technology, which is readily available as a product.

- Please provide information on the section deformation happening upon electron irradiation. Thin sections shrink and deform when first illuminated by electrons. This is especially true when applying the high flux that LaB6 filaments provide. How does this affect the bd acquisition? Does the beam for the central first tile already illuminate all neighboring tiles as well? Do you pre-irradiate the region to prevent shrinking? Do you correct for these effects post acquisition using non-rigid transformations when stitching the tiles?

We thank the reviewers for pointing out non-linear distortions that could exist in the raw image data. No pre-irradiation, except for a 50x overview image, was done before high-resolution imaging. The montages of 100 sections (see Data Availability) were successfully stitched with a proven software (alignTK) capable of compensating for non-linear deformations (e.g. Fig. 2e). The information and text has now been incorporated into the Results section (Lines 196 - 201).

- The software repository of pyAGTUM does not provide any documentation or installation instructions. Using a Leica ultramicrotome with an ATUM collector is of wide interest for the volume-EM community. An accessible solution (including the software) will be highly appreciated.

We regret not making better documentation of the software previously. The repository has now been fully populated with a general introduction, an explanation of key code, a 20-page manual with pictures, and a set up guide with a detailed list of hardware.

details:

- data availability: Is the source link correct? Pasting it into the source field results in an error while the direct hyperlink works.

To facilitate data sharing, we have now created a new landing page to avoid any errors associated with link pasting. Please see the Data Availability section.

Reviewer #1 (Remarks on code availability):

There is no sufficient documentation available to install or use the software.

Please see our response to the above request on improving the pyAGTUM code repository.

Reviewer #2 (Remarks to the Author):

The revised version did address many of the points raised previously, and the manuscript has been improved.

- However, the data is still not available to address at full resolution and only limited data is available. Can data be shared via IDR or BIA? Proper sharing is essential for review purposes as well as for the final readership of NC, if the paper will be accepted.

We apologized for the confusion. The data were shared at full resolution but loading time for high resolution images might vary. We have now created a new landing page, which will lead to the high resolution zoom level of the image data (see Data Availability).

- Suppl. F4 is redundant

The previous Supplementary Figure 4 has now been removed.

- Discuss resolution. Previously, this was incorrect (3nm pixsize = 3 nm resolution). Now it is no longer quantitative discussed.

We have included text regarding the axial resolution limitations in TEM imaging due to section thickness and the advantage of ion beam milling coupled with SEM imaging in this regard (Lines 272 - 279).

- I did not check all points, because access to the raw data at full resolution is crucial to give further comments

The EM data link in the new landing page should now lead to image data at full resolution. Please see the Data Availability section.

Reviewer #3 (Remarks to the Author):

The manuscript has clearly improved. Authors revised their manuscript to go beyond mere technical realization, but if main message is not on the technical realization, the purpose of this development is to generate useful data and thus to either have the data available to the community, or present further analyses based on that data. Insight into data and data quality has clearly improved, but I feel that a high-impact publication on a disruptive data generating pipeline should present that data in full to the community, and not a subset of sections.

While we share the reviewer's enthusiasm for publicizing more data, it is important to note that the successful acquisition, stitching, alignment, and reconstruction of a full EM dataset requires time and efforts that are beyond the scope of this study, and in the field was typically published as a major piece of work in itself. For example, imaging methods - Yin et al. Nat Commun 2020; alignment methods - Popovych et al. Nat Commun 2024; segmentation method - Januszewski et al. Nat Methods 2018; analysis of a dataset - Turner et al. Cell 2022. Therefore, we respectfully maintain that the stitched data of 100 sections at full resolution, each 1 mm² in size, are sufficient to demonstrate the feasibility and performance of the imaging pipeline.

The manuscript is now better embedded in the context of EM development, but I think relevance to cryo-EM and microbiology is far-fetched. VolumeEM incl connectomics and cellular biology and potentially extending to pathology in future is a very active, highly relevant field, no need to 'oversell'.

We thank the reviewer for recognizing the potential impact of this technology to related fields. The text in Discussion concerning cryo-EM and microbiology has been removed.

Abstract is now concise and stresses novelty, but the last sentence should be more specific for a broad audience: an array of how many in how many weeks?

We thank the reviewer for helping us improve the abstract. The number of microscopes and weeks has now been included.

There is a discrepancy in the comparison to previous literature: line 57 states prior state of the art achieved cubic millimeter in 6 months, while line 97 states authors achieve a two-fold increase in duty cycle.

We regretted not being clear with our descriptions previously. At least two factors account for the seemingly discrepancy. First, in the previous Line 57 six months include both imaging and non-imaging overheads such as stage movement and microscope downtime, whereas imaging duty cycle refers specifically to proportion of time spent on imaging, as defined in the preceding sentence. Second, six months were achieved with six microscopes (Yin et al. 2020), whereas imaging duty cycle is a per-microscope quantity. We have now revised the text to reflect nuances of the “6 months” claim (lines 61 - 63, 104 - 106).

The discussion and analysis of deflection and quality of deflected fields has improved, but the ray diagram added in Fig 1a is confusing. It seems there is a cross-over at the image plane, suggesting focused probe and no overlap with the deflected field. I think a better representation of the transmission system should be given with a (virtual) source magnified onto the sample plane and then projected onto an area on the camera. It also makes me wonder whether it is realistic that the beam still expands so much after the sample plane.

We have now improved the ray diagram and, to increase clarity, separate it into two panels. Please see the new Fig. 1a and legend.

Fig 2 is a great and needed addition to the paper. For Fig. 2d I think the y-axis is the spectral-SNR, with the SNR being the integral over the full spectral width. It would be nice to tabulate SNR's for the subtiles, or have this values in a panel 2e like in the inset in Fig 2d.

We are pleased that the addition of Fig 2 is appreciated. The y-axis label has been changed to spectral-SNR and a table of the SNR integrated over a frequency band is now included in Supplementary Fig. 1b.

All specific comments have been addressed.

Reviewer #4 (Remarks to the Author):

The authors have adequately addressed the reviewer's comments, and overall, the manuscript is well-crafted. However, I found some comments at this occasion with careful reading the manuscript. There is a challenge in understanding the effectiveness of integrating Cricket into the overall system, as well as the efficiency of increasing the number of tiling cards by Cricket from 9 to more than 9.

The authors have included a validation of the number of tiling sheets in the cricket, ensuring it exceeds 9. Although the evaluation is based solely on SNR, it is anticipated that distortion and focus blur would also increase. Would it not be more advantageous to provide a quantitative assessment of

these aspects as well? Furthermore, when considering a cricket with 9 tiling sheets, is it not essential to assess distortion and focus blur? Particularly, should these not be evaluated at the periphery of the diagonal image? Additionally, there might be a need for distortion correction; therefore, it would be beneficial to elaborate on the methodology employed for such correction.

We thank the reviewer for the constructive suggestion. Evaluation of distortion in a 9-tile configuration and the methodology has now been added to the manuscript (Fig. 2e, Results Lines 127 - 130, Methods lines 506 - 509). Overall the results show each sub-tile has a distinct and modest distortion field (maximum displacements of ~ 20 pixels in a 6k x 6k tile). Focus quality, reflected by sharpness of edges in images, contributes to our frequency domain-based SNR measurement. Notably, the fast Fourier transform-based algorithm was often used as a measurement of focus quality in large-scale imaging efforts. This discussion has now been added (lines 499 - 503).

It would be beneficial for readers if the authors could provide a comparative illustration for the following scenarios in a format consistent with Fig. 1d:

Without Cricket

With Cricket using 9 tiles

With Cricket using more than 9 tiles (e.g., 16 crickets)

A comparison of overheads between w/o Cricket, 9-tile, 16-tile has now been included in Supplementary Fig.1f. We have noted, in the manuscript, that performance increase beyond 9 tiles is limited given the proportion of stage movement overhead is already down to 4% of total time for 9 tiles (lines 102 - 104); this is also reflected in the figure.

REVIEWERS' COMMENTS

Reviewer #1 (Remarks to the Author):

The authors have properly addressed the comments raised previously. With little cosmetic changes, this manuscript is now well suited for publication.

l. 200: there seems to be a typo: "AlignTK"

Fig. 2a: please provide a scale bar

Fig. 2d: typo in axis caption

SF 4: typo in a: "buttons"

Reviewer #1 (Remarks on code availability):

The repository now contains documentation on the installation of hard and software. Documentation within the code itself can be improved.

Reviewer #2 (Remarks to the Author):

Two of my recurring points yet need to be addressed:

1. Data availability: In 2024 several options are present to share data in appropriate repositories as mentioned before. Github is not appropriate for data, it is for code. Why not IDR or BIA or ...? Key in large-scale EM is to share the data appropriately with the community.

2. Pixel size versus resolution: This is used incorrect. As this is key in EM/ the manuscript, it should be clearly explained what the parameters are/ how they relate. If field experts do not reach consensus, how can this be used/ interpreted by the final target audience?

Reviewer #3 (Remarks to the Author):

The authors have clarified the issues raised by the reviewers. The technical realization is now very clear and substantiated, the advancement in imaging capability well illustrated. I think this manuscript can be published as is, I only note minor (textual) corrections below.

p2 line 62-63 this is an awkward sentence, 2 in 1.

p5 line 201-202 can mentioned features be indicated in the images. The paper may be attractive to read for people in EM not in neuroscience or with a more technical background and they might not realize which features in the images is referred to.

p6 line 229 the SupFig that is referred to is not present anymore in the current manuscript

p6 line 257 'increase pixel size (i.e. microscope magnification)' better i.e. decrease magnification

Reviewer #4 (Remarks to the Author):

The authors have satisfactorily addressed the reviewer's previous comments, and overall, the manuscript demonstrates careful crafting. However, I encountered difficulty in comprehending figure 1a of the revised manuscript, which illustrates the ray path (ray diagram describing the optical path). The trajectory of the left ray without deflection by cricket appears notably distinct from the standard ray path typically employed in TEM imaging (e.g., as depicted in https://en.wikipedia.org/wiki/Transmission_electron_microscopy#/media/File:Schematic_view_of_imaging_and_diffraction_modes_in_TEM.tif). Are you suggesting that the ray path without beam deflection by cricket differs from the standard one used for imaging in TEM? If so, it would be beneficial for the readership if you could elucidate the distinctions and subsequently present the ray path when beam deflection by Cricket is applied.

REVIEWERS' COMMENTS

Reviewer #1 (Remarks to the Author):

The authors have properly addressed the comments raised previously. With little cosmetic changes, this manuscript is now well suited for publication.

I. 200: there seems to be a typo: "AlignTK"

Fig. 2a: please provide a scale bar

Fig. 2d: typo in axis caption

SF 4: typo in a: "buttons"

We thank the reviewers for pointing out to us. All of these corrections have now been made.

Reviewer #1 (Remarks on code availability):

The repository now contains documentation on the installation of hard and software. Documentation within the code itself can be improved.

We have now updated the repository with additional documentation within the code.

Reviewer #2 (Remarks to the Author):

Two of my recurring points yet need to be addressed:

1. Data availability: In 2024 several options are present to share data in appropriate repositories as mentioned before. Github is not appropriate for data, it is for code. Why not IDR or BIA or ...? Key in large-scale EM is to share the data appropriately with the community.

The size of our EM data, even with just 100 sections, amounts to 8 TB, which is larger than commonly accepted sizes of data repositories such as IDR.openmicroscopy and BiImage Archive. The landing page, hosted in Github, is set up as a gateway that leads to the image data for interactive viewing using the popular Neuroglancer program. The image data are hosted on our storage system and can be downloaded publicly using a widely-used, well-documented software developed by our lab (CloudVolume). This is done similarly to another recently published paper from our lab (Popovych et al. Nat. Commun. 15, 289. 2024). We have now updated the Data Availability section with clearer information on viewing and downloading the data.

Gateway webpage: <https://seung-lab.github.io/CA3Montaged>

Data address: https://s3-hpcrc.rc.princeton.edu/ca3-aligned-2023/image_cutout

CloudVolume: <https://github.com/seung-lab/cloud-volume>

2. Pixel size versus resolution: This is used incorrect. As this is key in EM/ the manuscript, it should be clearly explained what the parameters are/ how they relate. If field experts do not reach consensus, how can this be used/ interpreted by the final target audience?

All phrases of “pixel resolution” in the text have been corrected to “pixel size”. In addition, we have added a footnote in Table 1 to clarify this distinction.

“Pixel size is different from resolution, which refers to the capability of an imaging system to perceive two point sources separately.”

Reviewer #3 (Remarks to the Author):

The authors have clarified the issues raised by the reviewers. The technical realization is now very clear and substantiated, the advancement in imaging capability well illustrated. I think this manuscript can be published as is, I only note minor (textual) corrections below.

p2 line 62-63 this is an awkward sentence, 2 in 1.

Thanks. We have separated the sentence into 2.

p5 line 201-202 can mentioned features be indicated in the images. The paper may be attractive to read for people in EM not in neuroscience or with a more technical background and they might not realize which features in the images is referred to.

We have annotated Figure 3e and corresponding legends to indicate features in the image.

p6 line 229 the SupFig that is referred to is not present anymore in the current manuscript

Thanks. The SupFig reference has been removed.

p6 line 257 'increase pixel size (i.e. microscope magnification)' better i.e. decrease magnification

This phrase has been revised.

Reviewer #4 (Remarks to the Author):

The authors have satisfactorily addressed the reviewer's previous comments, and overall, the manuscript demonstrates careful crafting. However, I encountered difficulty in comprehending figure 1a of the revised manuscript, which illustrates the ray path (ray diagram describing the optical path). The trajectory of the left ray without deflection by cricket appears notably distinct from the standard ray path typically employed in TEM imaging (e.g., as depicted in https://en.wikipedia.org/wiki/Transmission_electron_microscopy#/media/File:Schematic_view_of_imaging_and_diffraction_modes_in_TEM.tif). Are you suggesting that the ray path without beam deflection by cricket differs from the standard one used for imaging in TEM? If so, it would

be beneficial for the readership if you could elucidate the distinctions and subsequently present the ray path when beam deflection by Cricket is applied.

We have now revised the Figure 1a ray diagram to conform to the common scheme.